# Effects of mango and mint pod-based e-cigarette aerosol inhalation on inflammatory states of the brain, lung, heart, and colon in mice

Alex Moshensky[1,2†], Cameron S Brand[3†], Hasan Alhaddad[4†], John Shin[1,2], Jorge A Masso-Silva[1,2], Ira Advani[1,2], Deepti Gunge[1,2], Aditi Sharma[5], Sagar Mehta[1,2], Arya Jahan[1,2], Sedtavut Nilaad[1,2], Jarod Olay[1,2], Wanjun Gu[2], Tatum Simonson[2], Daniyah Almarghalani[4], Josephine Pham[1,2], Samantha Perera[1,2], Kenneth Park[1,2], Rita Al-Kolla[1,2], Hoyoung Moon[3], Soumita Das[5], Min Kwang Byun[2,6], Zahoor Shah[7], Youssef Sari[4], Joan Heller Brown[3], Laura E Crotty Alexander[1,2*]

[1]Pulmonary and Critical Care Section, VA San Diego Healthcare System, La Jolla, United States; [2]Division of Pulmonary, Critical Care and Sleep Medicine and Section of Physiology, Department of Medicine, University of California San Diego (UCSD), San Diego, United States; [3]Department of Pharmacology, University of California San Diego (UCSD), San Diego, United States; [4]Department of Pharmacology and Experimental Therapeutics, College of Pharmacy and Pharmaceutical Sciences, University of Toledo, Toledo, United States; [5]Department of Pathology, University of California San Diego (UCSD), San Diego, United States; [6]Division of Pulmonology, Department of Internal Medicine, Gangnam Severance Hospital, Yonsei University College of Medicine, Seoul, Republic of Korea; [7]Department of Medicinal and Biological Chemistry, College of Pharmacy and Pharmaceutical Sciences, University of Toledo, Toledo, United States

*For correspondence:
lca@ucsd.edu

†These authors contributed equally to this work

Competing interest: The authors declare that no competing interests exist.

**Abstract** While health effects of conventional tobacco are well defined, data on vaping devices, including one of the most popular e-cigarettes which have high nicotine levels, are less established. Prior acute e-cigarette studies have demonstrated inflammatory and cardiopulmonary physiology changes while chronic studies have demonstrated extra-pulmonary effects, including neurotransmitter alterations in reward pathways. In this study we investigated the impact of inhalation of aerosols produced from pod-based, flavored e-cigarettes (JUUL) aerosols three times daily for 3 months on inflammatory markers in the brain, lung, heart, and colon. JUUL aerosol exposure induced upregulation of cytokine and chemokine gene expression and increased HMGB1 and RAGE in the nucleus accumbens in the central nervous system. Inflammatory gene expression increased in the colon, while gene expression was more broadly altered by e-cigarette aerosol inhalation in the lung. Cardiopulmonary inflammatory responses to acute lung injury with lipopolysaccharide were exacerbated in the heart. Flavor-specific findings were detected across these studies. Our findings suggest that daily e-cigarette use may cause neuroinflammation, which may contribute to behavioral changes and mood disorders. In addition, e-cigarette use may cause gut inflammation, which has been tied to poor systemic health, and cardiac inflammation, which leads to cardiovascular disease.

## Editor's evaluation

This study aimed to investigate the effects of vaping on inflammatory cytokine expression in multiple organs in mice. The effects of e-cigarette exposures on a different organs remain highly under investigated and, thus this study is of high interest. The manuscript has significantly improved and the authors added additional data to support their conclusion and further adapted the manuscript text and highlighted the limitations of this study.

## Introduction

Chronic inhalation of tobacco smoke is known to damage multiple cell types and cause a wide range of diseases throughout the body. In particular, many pulmonary inflammatory diseases are caused and affected by conventional tobacco use (*Shin and Crotty Alexander, 2016*; *Crotty Alexander et al., 2015a*). It is also known that nicotine affects brain development and alters responses to addictive substances. Nicotine activates carcinogenic pathways, putting users at an increased risk of cancer (*Services USDoHaH, 2014*). With unproven claims to be a safer alternative to cigarette smoking, modern electronic (e)-cigarette devices were introduced in 2003 as a novel nicotine delivery system (*Cahn and Siegel, 2011*; *Crotty Alexander et al., 2015b*). The JUUL, a device that gained popularity due to its sleek and concealable design, has utilized pods containing e-liquids with enticing flavors such as Mango, Mint, and Crème Brulee (now discontinued) (*Huang et al., 2019*). However, the health effects of chronic inhalation of aerosols generated from pod devices remain largely unknown.

While the data on health effects of conventional tobacco are extensive, the data on e-cigarettes and vaping devices are less established due to their recent entry to the market (*Tsai et al., 2020*; *Bozier et al., 2020*). In particular, research in this area is impeded by the rapid evolution of vaping devices. The vape pens and cig-a-likes were the first e-cigarettes studied from 2007 to 2014, whereas the box Mods became highly popular and research on these devices began around 2015. Pod devices, including the JUUL, were invented in 2016 and rapidly dominated the market by 2017–2020 (*Craver, 2019*). These pod-based devices produce aerosols with a different chemical composition than prior devices, including often significant higher concentrations of nicotine than Mod devices. Studies of JUUL to date have been predominantly subacute and acute exposures with a focus on in vitro and in vivo experiments. A majority of the current literature examined the cytotoxic effects of varying JUUL flavors (Crème Brulee, Cool Mint, Fruit Medley, Tobacco, Menthol, etc.) on cells of the respiratory system (*Pinkston et al., 2020*; *Ramirez et al., 2020*; *Pearce et al., 2020*; *Muthumalage et al., 2019*; *O'Farrell et al., 2021*; *Rao et al., 2020*; *Ghosh et al., 2021*; *Lamb et al., 2020*). Most in vitro studies concluded that JUUL aerosols are cytotoxic and impair cell function (*Pinkston et al., 2020*; *Pearce et al., 2020*; *Muthumalage et al., 2019*; *O'Farrell et al., 2021*; *Rao et al., 2020*; *Ghosh et al., 2021*). Additional studies defined the chemical profiles of JUUL flavored cartridges (*Ramirez et al., 2020*) or examined aerosol emission and oxidant yields from flavored JUUL pods (*Talih et al., 2019*; *Reilly et al., 2019*). These studies concluded that while JUUL pods had significantly lower oxidant yields in comparison to combustible cigarettes, the nicotine concentrations were substantially greater (*Talih et al., 2019*; *Reilly et al., 2019*; *Omaiye et al., 2019*). Another study assessed whether exposure to JUUL aerosol (in comparison to aerosol containing Vitamin E Acetate) played a causal role in EVALI patients (*Matsumoto et al., 2020*). While the study revealed that 15 days of exposure to JUUL aerosol did not cause direct lung injury, the authors cautioned that chronic exposure to nicotine may still have a disruptive effect on lung physiology (*Matsumoto et al., 2020*). While these studies have laid the groundwork for assessing the acute impact of JUUL use on the respiratory system, much work remains to be done on the effects across the body.

Because of the short time e-cigarettes and vaping devices have been on the market, very little is known about the long-term effects of vaping. Acute and subacute studies (days to weeks) in human subjects have demonstrated changes in lung and cardiac function, with increased airway reactivity and lung inflammation, and increased heart rate and blood pressure in response to vaping (*Tsai et al., 2020*). Previous studies of chronic effects of vaping are limited to e-cigarette aerosol inhalation models in animals (chronic defined as ≥3 months of daily exposure) but have demonstrated more profound effects, including renal, cardiac, and liver fibrosis (*Crotty Alexander et al., 2018*), emphysema (*Garcia-Arcos et al., 2016*), lung cancer (*Tang et al., 2019*), increased lung injury in the setting of influenza infection (*Madison et al., 2019*), increased arterial stiffness and atherosclerosis,

**eLife digest** The use of e-cigarettes or 'vaping' has become widespread, particularly among young people and smokers trying to quit. One of the most popular e-cigarette brands is JUUL, which offers appealing flavors and a discrete design. Many e-cigarette users believe these products are healthier than traditional tobacco products. And while the harms of conventional tobacco products have been extensively researched, the short- and long-term health effects of e-cigarettes have not been well studied. There is even less information about the health impacts of newer products like JUUL.

E-cigarettes made by JUUL are different relative to prior generations of e-cigarettes. The JUUL device uses disposable pods filled with nicotinic salts instead of nicotine. One JUUL pod contains as much nicotine as an entire pack of cigarettes (41.3 mg). These differences make studying the health effects of this product particularly important.

Moshensky, Brand, Alhaddad et al. show that daily exposure to JUUL aerosols increases the expression of genes encoding inflammatory molecules in the brain, lung, heart and colon of mice. In the experiments, mice were exposed to JUUL mint and JUUL mango flavored aerosols for 20 minutes, 3 times a day, and for 4 and 12 weeks. The changes in inflammatory gene expression varied depending on the flavor. This suggests that the flavorings themselves contribute to the observed changes.

The findings suggest that daily use of pod-based e-cigarettes or e-cigarettes containing high levels of nicotinic salts over months to years, may cause inflammation in various organs, increasing the risk of disease and poor health. This information may help individuals, clinicians and policymakers make more informed decisions about e-cigarettes. Further studies assessing the impact of these changes on long-term physical and mental health in humans are desperately needed. These should assess health effects across different e-cigarette types, flavors and duration of use.

and activation of addiction neurocircuits in the brain (*Alasmari et al., 2017*; *Linker et al., 2020*). Overall, the health effects of vaping JUUL pods remain unknown, despite the popularity of pod-based e-devices.

Here, we broadly assessed the effects of daily JUUL aerosol inhalation on cardiopulmonary function and inflammation across organ systems, including the reward pathways in the brain. We also induced lung injury to determine whether chronic JUUL use predisposes to deleterious responses in the setting of common infectious challenges. The findings presented here raise the concern that daily inhalation of JUUL aerosols may alter inflammation in the brain, heart, lung, and colon, as well as alter physiologic functions.

## Results

### Daily JUUL inhalation for 3 months is associated with neuroinflammation

Previous studies have shown that conventional tobacco smoking increases proinflammatory cytokines in the brain, specifically *Tnfa*, *Il1b*, *Il6* (*Lau et al., 2012*; *Bradford et al., 2011*). Therefore, gene expression of these inflammatory cytokines were measured by qPCR in different brain regions of mice exposed to JUUL Mango and JUUL Mint, as well as Air controls for one or 3 months. Specifically, we assessed gene expression in the nucleus accumbens core and shell (NAc-core and NAc-shell), and hippocampus, regions involved in behavior modification, formation of drug reward and anxious or depressive behaviors, and learning and memory, respectively. We observed that *Tnfa* gene expression was significantly increased in the NAc-core and NAc-shell of mice exposed to one or 3 months of JUUL Mango or JUUL Mint compared to Air controls (*Figure 1A–D*). In contrast, *Tnfa* levels were unchanged in the hippocampus throughout the exposures (*Figure 1E–F*). *Il1b* gene expression was also significantly elevated in JUUL Mint- and Mango-exposed mice in both the NAc-core and NAc-shell at 1 month compared to air controls (*Figure 1G and I*) but remained elevated at 3 months only in the NAc-shell (*Figure 1J*). The hippocampus showed unchanged levels of *Il1b* gene expression at 1 and 3 months across groups (*Figure 1K and L*). In the case of *Il6*, we observed a significant increase in gene expression in the NAc-shell in both JUUL Mango and JUUL Mint groups at 1 and 3 months

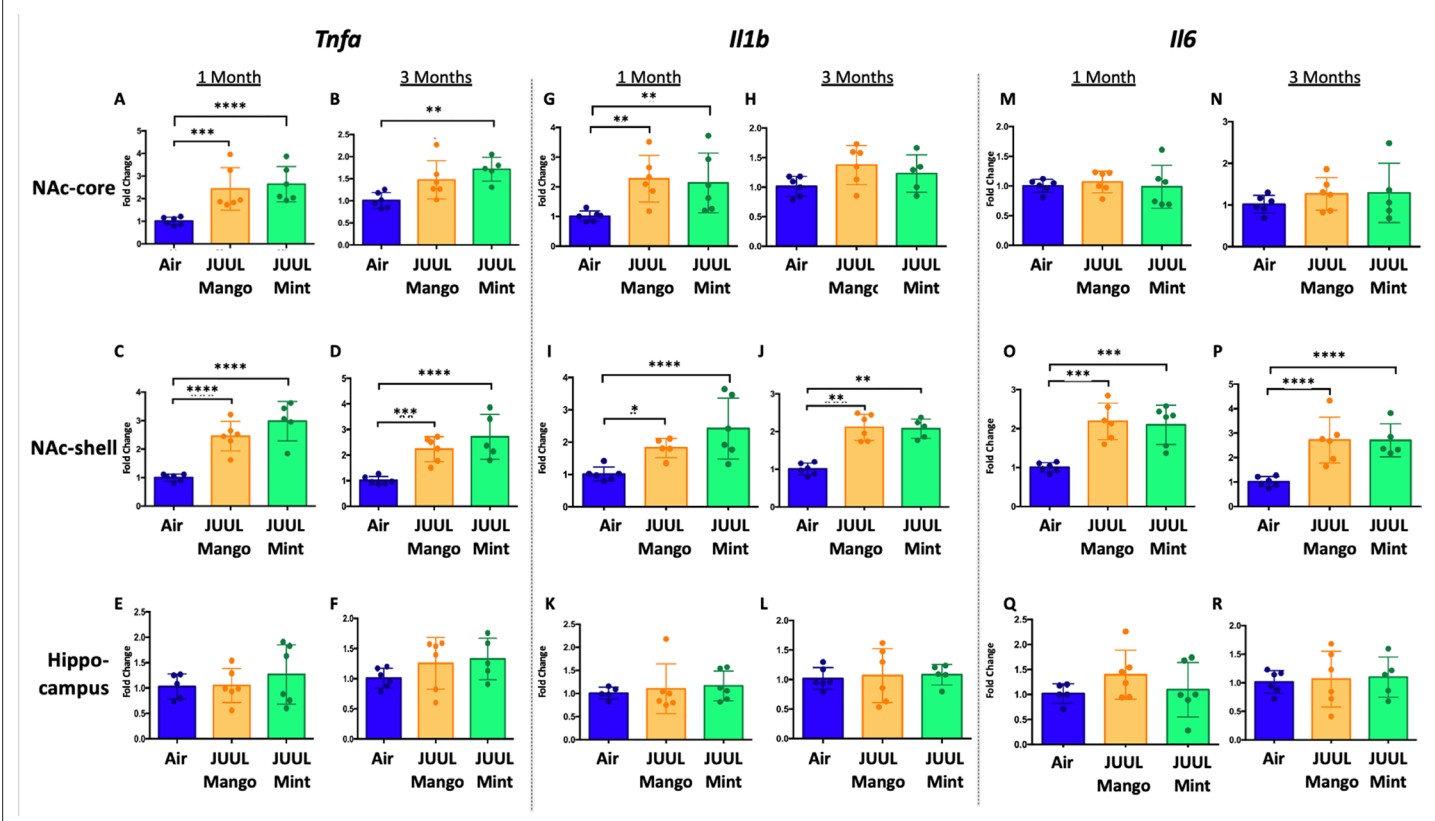

**Figure 1.** Three months of JUUL aerosol inhalation leads to an increase of pro-inflammatory cytokines in different regions of the brain. Brains were harvested at the end point and the regions for NAc-core, NAc-shell and Hippocampus were harvested and frozen. RNA was extracted and qPCR was performed to quantify the expression of *Tnfa, Il1b, Il6*. *Tnfa* expression is shown from NAc-core at (**A**) 1 month and (**B**) 3 months, from NAc-shell at (**C**) 1 month and (**D**) 3 months, and from Hippocampus at (**E**) 1 month and (**F**) 3 months. *Il1b* expression is shown from NAc-core at (**G**) 1 month and (**H**) 3 months, from NAc-shell at (**I**) 1 month and (**J**) 3 months, and from Hippocampus at (**K**) 1 month and (**L**) 3 months. *Il6* expression is shown from NAc-core at (**M**) 1 month and (**N**) 3 months, from NAc-shell at (**O**) 1 month and (**P**) 3 months, and from Hippocampus at (**Q**) 1 month and (**R**) 3 months. Data were analyzed with two-way ANOVA with Dunnett's multiple comparisons for each brain region and timepoint. Data are presented as individual data points ± SEM with n = 5–6 mice per group. *p < 0.05, **p < 0.01, *** p < 0.001 and **** p < 0.0001.

The online version of this article includes the following figure supplement(s) for figure 1:

**Figure supplement 1.** Inflammatory gene expression changes in the central nervous system in mice exposed to JUUL Mango and JUUL Mint.

(*Figure 1O and P*), but no significant differences were observed in the NAc-core and hippocampus when compared to Air controls (*Figure 1M, N, Q and R*; *Figure 1—figure supplement 1*). Overall, these data suggest that exposure to JUUL Mint and JUUL Mango may induceneuroinflammation in brain regions responsible for behavior modification, drug reward, and formation of anxious or depressive behaviors (*Russo and Nestler, 2013*).

To further confirm the neuroinflammatory response associated with chronic JUUL exposure, we measured protein levels of receptors for advanced glycation end products (RAGE) and its ligand high mobility group Box 1 (HMGB1) protein by western blot in the NAc-core, NAc-shell, and hippocampus of mice exposed to JUUL Mango, JUUL Mint and Air at 1 and 3 months. RAGE and HMGB1 have been implicated in inducing neuroinflammation (*Ghosh et al., 2021*), and previous studies have shown that HMBG1-1 and RAGE expression are increased with exposure to cigarette smoke (*Le et al., 2020*; *Robinson et al., 2012*). No significant changes of HMGB1 were observed in NAc-core at 1 or 3 months of JUUL aerosol exposurebetween groups (*Figure 2A–B*). The NAc-shell, however, showed significant increase in HMGB1 at 1 and 3 months in mice exposed to JUUL Mango and JUUL Mint relative to Air controls (*Figure 2C–D*), and the increase was more pronounced at 3 months (*Figure 2D*). The hippocampus showed no changes in HMGB1protein expression at 1 month (*Figure 2E*), and actually showed significant decrease in protein expression in mice exposed for 3 months to either

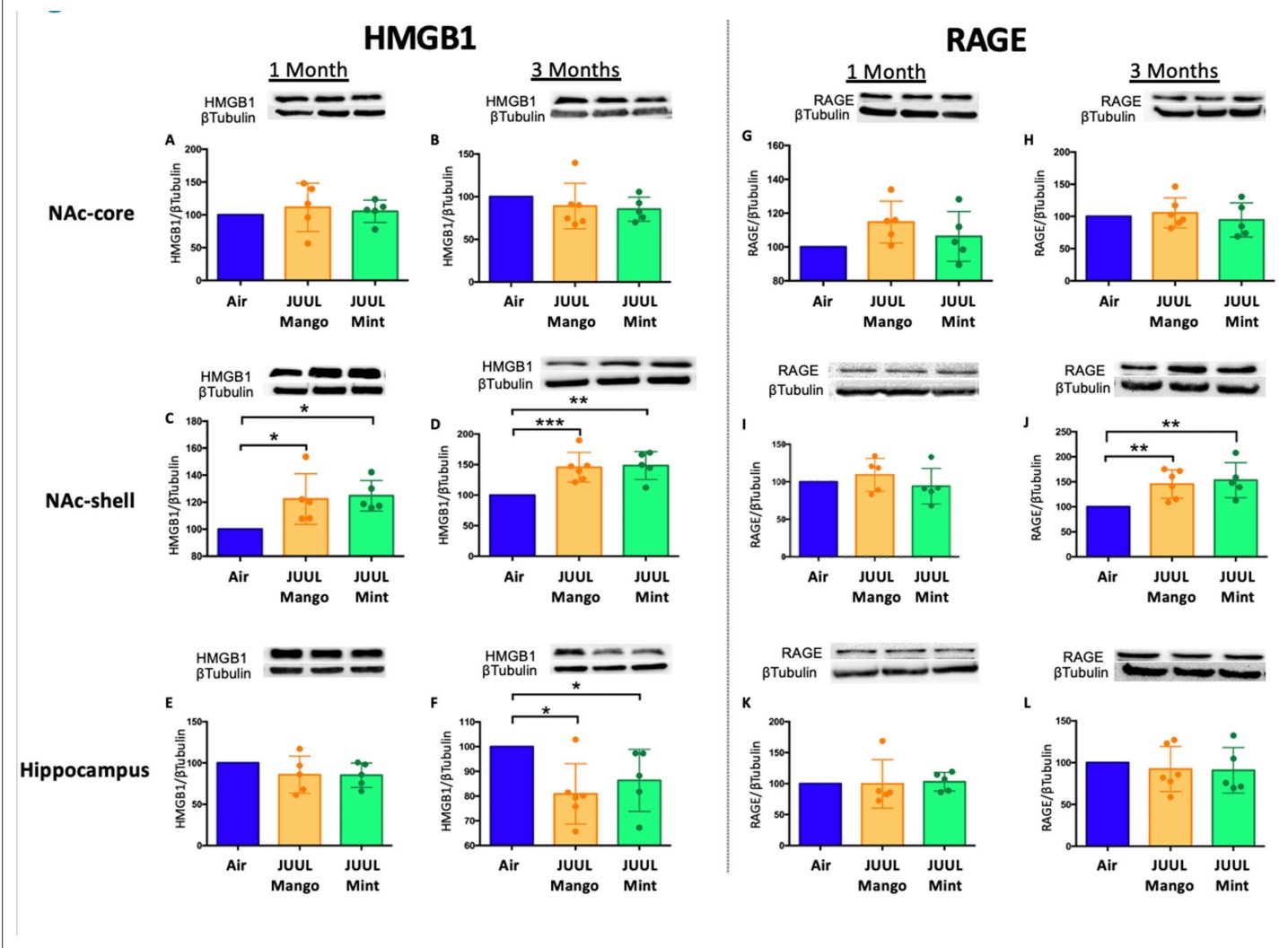

**Figure 2.** Three months of JUUL aerosol inhalation leads to an increase of inflammatory mediators HMGB1 and RAGE. Brains were harvested at the end point and the regions for NAc-core, NAc-shell and Hippocampus were sectioned. Later, protein was extracted and Western Blot was performed to quantify the expression of HMGB1-1 and RAGE. HMGB1-1 relative protein level are shown from NAc-core at (**A**) 1 month and (**B**) 3 months, from NAc-shell at (**C**) 1 month and (**D**) 3 months, and from Hippocampus at (**E**) 1 month and (**F**) 3 months. RAGE protein levels are shown from NAc-core at (**G**) 1 month and (**H**) 3 months, from NAc-shell at (**I**) 1 month and (**J**) 3 months, and from Hippocampus at (**K**) 1 month and (**L**) 3 months. Changes in proteins levels are relative to Air controls. Data are presented as individual data points $\pm$ SEM with n = 5–6 mice per group. *p < 0.05, **p < 0.01 and *** p < 0.001.

The online version of this article includes the following figure supplement(s) for figure 2:

**Figure supplement 1.** Inflammatory protein level changes in the central nervous system in mice exposed to JUUL Mango and JUUL Mint.

JUUL Mango or JUUL Mint as compared to Air controls (*Figure 2F*). In the case of RAGE, the protein levels were not significantly altered in all tested brain regions (*Figure 2G–I and K–L*), except in the NAc-shell of the mice exposed to JUUL Mango or JUUL Mint for 3 months when compared to Air controls (*Figure 2J*; *Figure 2—figure supplement 1*). Altogether, these data suggest that exposure to aerosols from JUUL devices induced neuroinflammation in reward brain regions, particularly that of the NAc-shell and NAc-core regions.

## Inhalation of JUUL aerosols for 3 months alters inflammatory state and fibrosis associated gene expression in cardiac tissue

Changes in the myocardium have been widely observed in response to cigarette smoking, and we previously showed that inhalation of e-cigarette aerosols generated by second generation e-cigarettes

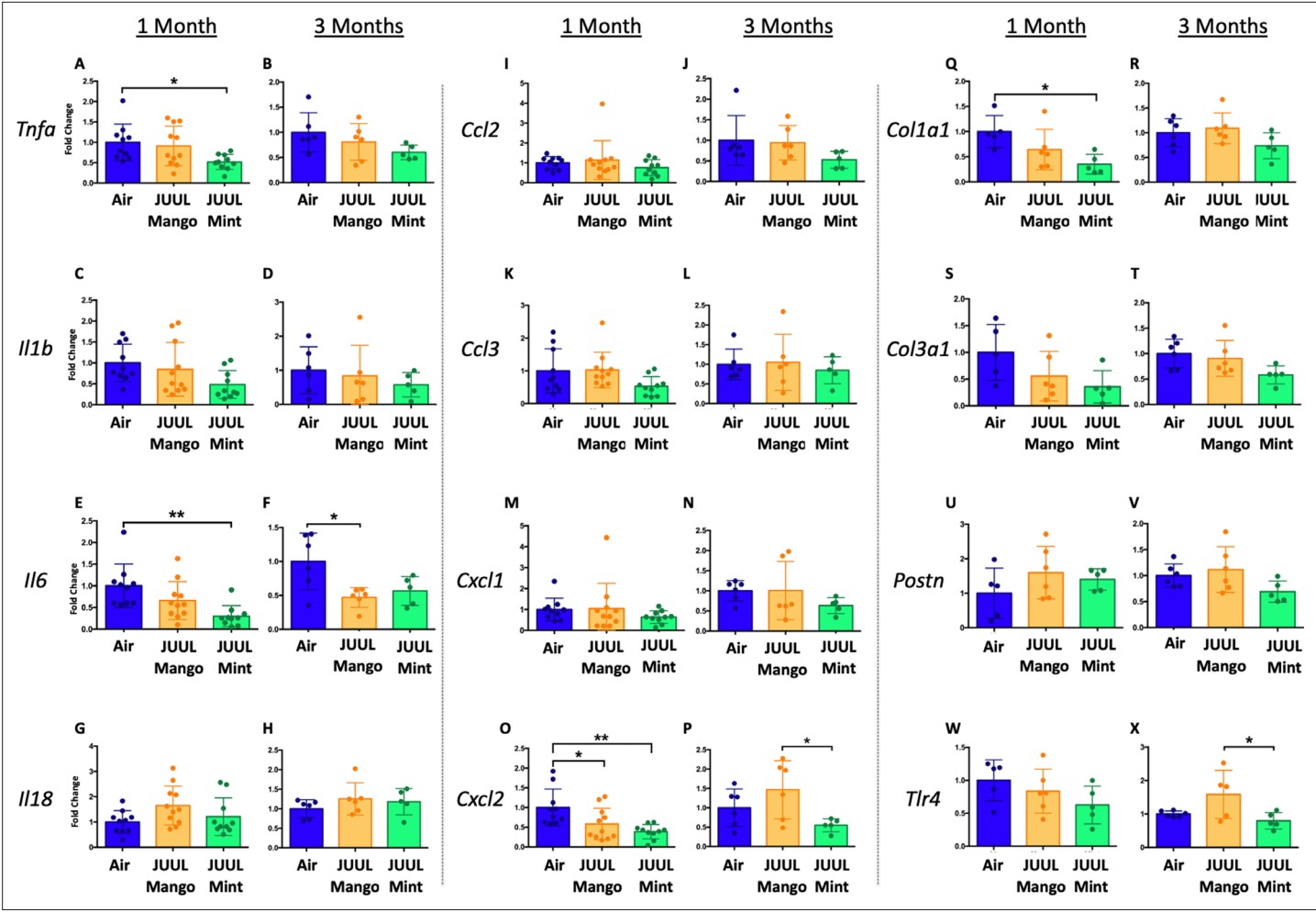

**Figure 3.** Three months of inhalation of JUUL aerosols alters inflammatory and fibrosis associated gene expression in cardiac tissue. Hearts were harvested, and RNA was extracted from the left ventricle and qPCR was performed to quantify the gene expression of different cytokines, chemokines and fibrosis-associated genes. Cytokines include *Tnfa* at (**A**) 1 month and (**B**) 3 months, *Il1b* at (**C**) 1 month and (**D**) 3 months, *Il6* at (**E**) 1 month and (**F**) 3 months, and *Il18* at (**G**) 1 month and (**H**) 3 months. Chemokines include *Ccl2* at (**I**) 1 month and (**J**) 3 months, *Ccl3* at (**K**) 1 month and (**L**) 3 months, *Cxcl1* at (**M**) 1 month and (**N**) 3 months, and *Cxcl2* at (**O**) 1 month and (**P**) 3 months. Fibrosis-associated genes include *Col1a1* at (**Q**) 1 month and (**R**) 3 months, *Col3a1* at (**S**) 1 month and (**T**) 3 months, *Postn* at (**U**) 1 month and (**V**) 3 months, and *Tlr4* at (**W**) 1 month and (**X**) 3 months. Changes in expression levels are relative to Air controls. Data are presented as individual data points ± SEM with n = 5–11 mice per group. *p < 0.05 and **p < 0.01.

The online version of this article includes the following figure supplement(s) for figure 3:

**Figure supplement 1.** Inflammatory gene expression changes in the hearts of mice exposed to JUUL Mango and JUUL Mint.

(vape pens) for 3–6 months induced fibrotic changes in cardiac tissue (*Crotty Alexander et al., 2018*). Fibrosis is typically driven by either cellular injury or inflammation. Increases in pro-inflammatory cytokines and fibrosis-associated proteins have been linked to the development of cardiovascular diseases (*Pearson et al., 2003*; *Vasan et al., 2003*; *Liu et al., 2017*; *Ma et al., 2018*). Based on the inflammatory and fibrotic markers commonly observed in response to myocardial infarction and development of heart failure, we assessed the expression of mRNA for *Tnfa, Il1b,Il6, Il18, Ccl2, Ccl3, Cxcl1, Cxcl2, Col1a1, Col3a1, Postn,* and *Tlr4* at 1 and 3 months (*Figure 3A–X*). Surprisingly, none of the pro-inflammatory cytokines or chemokines examined were upregulated by JUUL exposure. Indeed, *Tnfa, Il6,* and *Cxcl2* were downregulated in 1 month JUUL Mint exposed mice, as was the pro-fibrotic gene *Col1a1* (*Figure 3A, E, O and Q*). In contrast to JUUL Mint, aerosol inhalation of JUUL Mango for 1 month was only associated with downregulation of *Cxcl2* (*Figure 3O*). JUUL Mint and Mango aerosol inhalation also had differing effects on *Cxcl2* and *Tlr4* expression at 3 months of exposure (*Figure 3P and X,* respectively; *Figure 3—figure supplement 1*). These findings suggest that there may be flavor-specific effects in addition to nicotine-specific effects on the cardiovascular system.

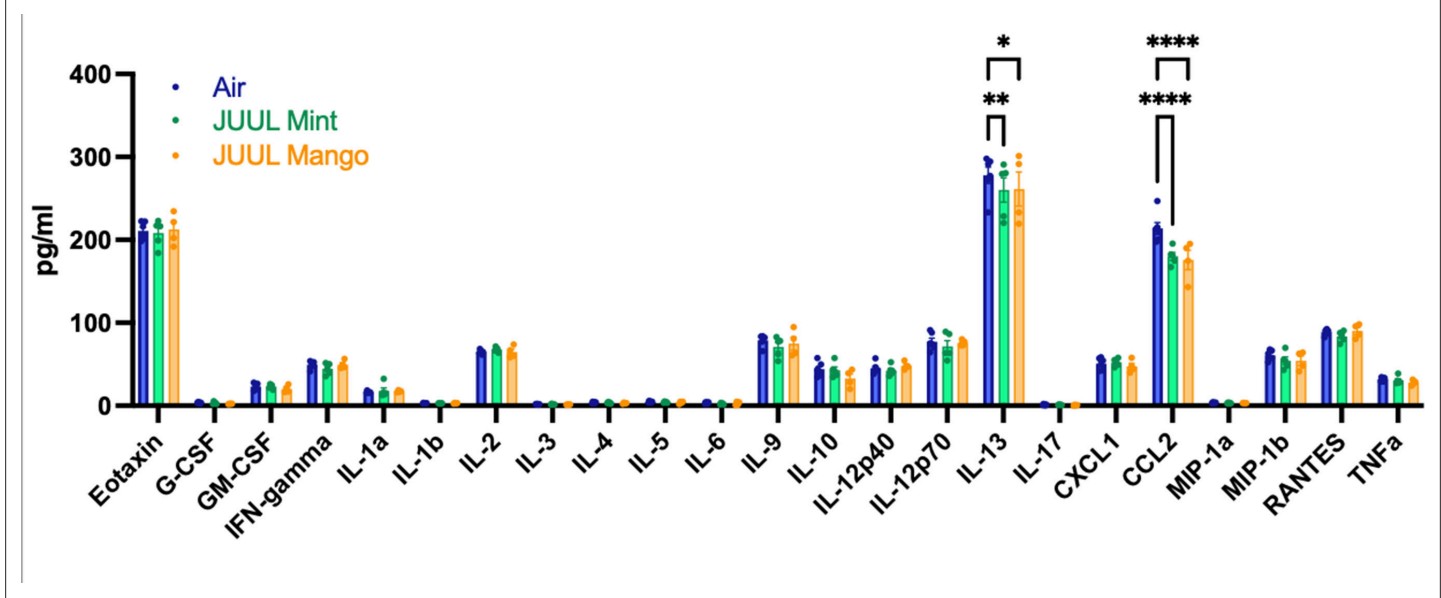

**Figure 4.** Chronic inhalation of JUUL aerosols alters *Ccl2* and *Il13* levels in cardiac tissue. Cardiac apex tissue was lysed, total protein isolated, and inflammatory proteins quantified by Bio-Plex Pro Mouse Cytokine 23-plex Assay. Both Il13 and Ccl2 levels were diminished in cardiac tissue from mice exposed for 3 months to JUUL Mint (green) and JUUL Mango (orange) aerosols, relative to Air controls (blue). Data was analyzed by two-way ANOVA with Dunnett's corrections for multiple comparisons, and are presented as individual data points ± SEM, with n = 4–6 mice per group. *p = 0.017, **p < 0.01, and ****p < 0.0001.

Upon broad multiplex assessment for protein-level based changes in the myocardium, levels *ofCcl2* were significantly decreased in the cardiac apex of mice exposed to JUUL Mint and JUUL Mango for 3 months (*Figure 4*). *Il13* was reduced to a lesser degree in the cardiac tissue of both JUUL Mint and JUUL Mango exposed mice (*Figure 4*). Overall, these changes in gene expression and inflammatory proteins (Air vs JUUL Mint p < 0.01 and Air vs JUUL Mango p < 0.001, Friedman test with Dunn's multiple comparisons)suggest that inflammatory pathways in cardiac tissue may be affected by JUUL aerosol inhalation. While overt inflammation is not apparent, it is well known that any alterations to the immune-inflammation axis, activating or suppressive, can lead to changes in disease susceptibility and incidence (*Bennett et al., 2018*).

## Inhalation of JUUL aerosols for 3 months alters pro-inflammatory markers in the colon

Because cigarette smoking has been shown to alter inflammation in the gut and promote chronic digestive diseases (*Verschuere et al., 2012*; *Berkowitz et al., 2018*), and because e-cigarette aerosols also deposit in the oro- and retro- pharynx, leading to introduction of the inhaled chemicalsinto the gastrointestinal (GI) tract, we evaluated the effects of JUUL aerosol inhalation on the GI tract. In terms of documenting the effects of e-cigarettes on GI inflammation, our knowledge is limited to only the study done by our research group, with a focus on changes induced by aerosols generated by third-generation e-cigarettes (box Mods) only (*Sharma et al., 2020*). In order to assess JUUL induced changes in the GI tract, we examined inflammatory gene expression in the colon at 1 and 3 months of JUUL exposure. JUUL Mango induced upregulation of *Tnfa*, *Il6*, and *Il8* relative to Air controls after 1 month exposure (*Figure 5A, C and G*). Interestingly, at 3 months, JUUL Mango treatment resulted in less expression of *Tnfa*, *Il6* and *Il1b* than that observed in Air controls or JUUL Mint exposed mice (*Figure 5B, D and F*), but increased expression of *Ccl2* (*Figure 5J*). These data suggest that exposure to JUUL Mango aerosols modulates inflammation in the colon, with induction of key inflammatory cytokines at 1 month (sub-acute exposure). In JUUL Mango and Mint, there was no change in *Il1b* or *Ccl2* at 1 month (*Figure 5E and I*) and *Il8* at 3 months (*Figure 5H*; *Figure 5— figure supplement 1*).

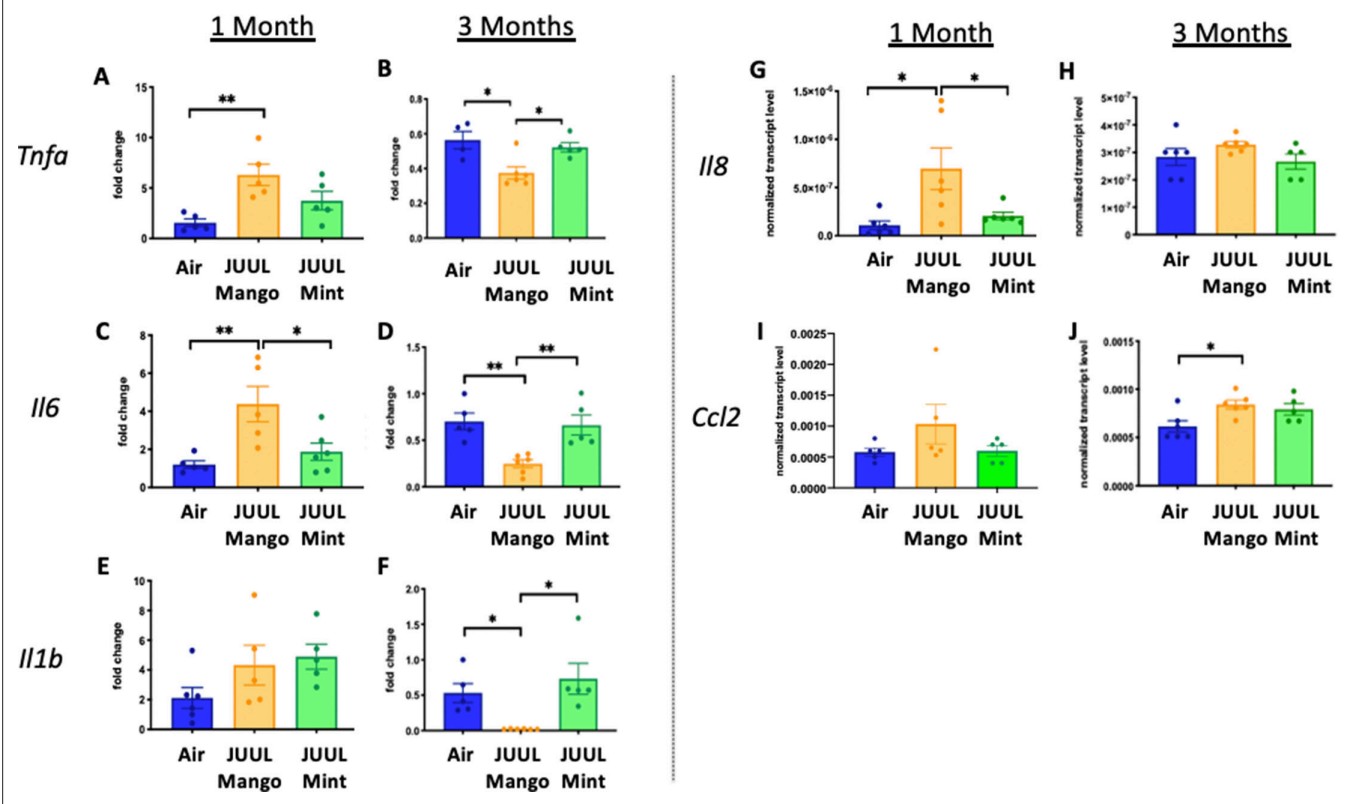

**Figure 5.** Three months of JUUL aerosol inhalation alters pro-inflammatory markers in colon. Inflammation was assessed in the colon at 1 and 3 months. Panels show inflammation markers in the colon in *Tnfa* (**A**) 1 month and (**B**) 3 months, *Il6* at (**C**) 1 month and (**D**) 3 months, *Il1b* at (**E**) 1 month and (**F**) 3 months,*Il8* (**G**) and (**H**), and *Ccl2*l 1 month and (**J**) 3 months. Data for inflammation markers is presented as individual data points ± SEM. * p < 0.01 and ** p < 0.001.

The online version of this article includes the following figure supplement(s) for figure 5:

**Figure supplement 1.** Inflammatory gene expression changes in the colon of mice exposed to JUUL Mango and JUUL Mint.

## Daily JUUL aerosol inhalation does not alter cardiopulmonary physiology

Chronic exposure to cigarette smoke leads to cardiovascular changes, mediated through altered autonomic tone, but little is known about the chronic cardiovascular effects of e-cigarettes, especially with fourth generation (pod) devices (*Tsai et al., 2020*). Thus, we exposed mice to JUUL aerosols and carried out assessments of cardiovascular function, including blood pressure (BP), heart rate (HR), and heart rate variability (HRV). Heart rate variability was determined from root-mean square differences of successive R-R intervals (RMSSD) and the mean of the standard deviations for all R-R intervals (SDNN). There were no significant changes in HR or HRV at 1 and 3 months of either JUUL Mint or JUUL Mango exposure relative to Air controls (*Appendix 1—figure 1*). Similarly, systolic and diastolic BP were also unchanged relative to Air controls at either one or 3 months (*Appendix 1—figure 1*). Thus, chronic exposure to pod-based e-cigarette aerosols containing high levels of nicotine may not alter normal physiological autonomic cardiovascular regulation in mice.

Lungs represent the main site for aerosol deposition during inhalant use. Several studies have shown the effects of e-cigarettes on lung physiology (*Tsai et al., 2020*; *Glynos et al., 2018*). To determine the effect of chronic JUUL aerosol inhalation on lung physiology, lung function studies including mechanic scans of airways resistance and lung elastance were carried out and were found to be similar across JUUL Mint, JUUL Mango and Air control groups at 1 and 3 months of exposure (*Appendix 1—figure 2*). Pressure-volume (PV) loops also demonstrated similarities amongst the three groups at 1 and 3 months (*Appendix 1—figure 2*). Airway hyperreactivity was tested by methacholine challenge and revealed no differences amongst groups, as measured at 1 and 3 months (*Appendix 1—figure 2*).

Thus, 1 and 3-month exposure to JUUL Mint and Mango aerosols may not cause significant changes in airway physiology, but this does not preclude the possibility that changes may occur with longer exposures, such as 6–12 months.

## JUUL aerosol inhalation for 3 months alters the inflammatory state of lungs only at the level of gene expression under homeostatic conditions

Conventional tobacco as well as some e-cigarette aerosol exposures, have been found to cause increased cellularity in the airways (*Crotty Alexander et al., 2015a*; *Tsai et al., 2020*; *Bozier et al., 2020*), and cigarette smoking leads to recruitment of neutrophils to the airways in particular (*Crotty Alexander et al., 2015a*). Total leukocyte and neutrophil counts in bronchoalveolar lavage (BAL) of mice exposed to JUUL aerosols for 1 and 3 months were no different than those in Air controls, indicating that inflammatory cell recruitment to the airways was unaffected (*Appendix 1—figure 3*). Moreover, fixed lung sections stained with H&E showed no difference in inflammation in the lungs at 1 and 3 months of JUUL aerosol exposure as compared to Air controls (*Appendix 1—figure 4A–L*). It is well known that cigarette smoke inhalation leads to emphysematous changes in the lungs of chronic smokers, and some in vivo murine studies have suggested that e-cigarette vaping may also cause emphysema. Mean linear interface (MLI) assessments on lungs yielded no detection of increased airspaces in mice exposed to JUUL Mint or JUUL Mango aerosols for one or 3 months (*Appendix 1—figure 4M*).

While evidence of inflammation at the cellular level, by increased numbers of cells within the airways and parenchyma,is overt proof of inflammation and immunomodulation occurring in response to inhalation of chemicals within e-cigarette aerosols, alterations in the phenotype of cells throughout the lungs can be detected by transcriptomics, and can identify covert pathology which may only become apparent over a long period of time or in the setting of inflammatory, infectious or toxic challenges.Broad assessment of gene expression with RNAseq identified vaping induced changes not identifiable at the cellular, tissue and physiology levels within the lungs of mice that inhaled JUUL aerosols. Exposure to Mango JUUL aerosols led to 155 significant gene expression changes, while Mint JUUL aerosols led to 74 (*Figure 6A*). These gene expression changes are most likely to be due to the chemicals added to the e-liquids to create the Mint and Mango flavors, in contrast to the 99 gene expression changes that were found to be common across the two JUUL exposures and likely due to the chemicals common to both Mint and Mango JUUL pods (nicotinic salts, propylene glycol, glycerol and benzoic acid, among others). The genes whose expression was most significantly altered included GTPases (nicotine), mucins (mint flavor), CCL6 (mango flavor), and TGFb receptors (nicotine and mango flavor; *Figure 6B*).

By looking at volcano plots of gene expression changes specific for each inhalant, at each time-point, we found that exposure to JUUL Mint aerosols led to increased changes over time, while effects of JUUL Mango remained relatively stable (*Figure 6C*). Furthermore, upon challenge with inhaled LPS, mice whichhad been chronically exposed to JUUL Mint aerosols had a large number of gene expression changes that were disparate from those of Air controls and JUUL Mango exposed mice (*Figure 6D*).

## JUUL aerosol inhalation for 3 months does not induce cardiac, renal, or liver fibrosis

Our previous studies with mice exposed to aerosols generated from Vape pens not only found fibrosis in cardiac tissue after 3–6 months of e-cigarette aerosol inhalation but also in the liver and kidneys (*Crotty Alexander et al., 2018*). Cigarette smoking is also known to cause organ fibrosis (*Drummond et al., 2016*). There were, however, no significant changes in fibrosis assessed by quantification of collagen fibers stained with Masson's trichrome, in the liver, heart or kidneys of mice that inhaled JUUL Mango or JUUL Mint aerosols for 3 months relative to Air controls (*Appendix 1—figure 5*).

## Impact of JUUL exposure on airway inflammation in the setting of inhaled LPS challenge

Long-term cigarette smoking is known to predispose to greater inflammatory responses to lung infections (*Crotty Alexander et al., 2015a*). However, few studies have examined the effects of e-cigarette

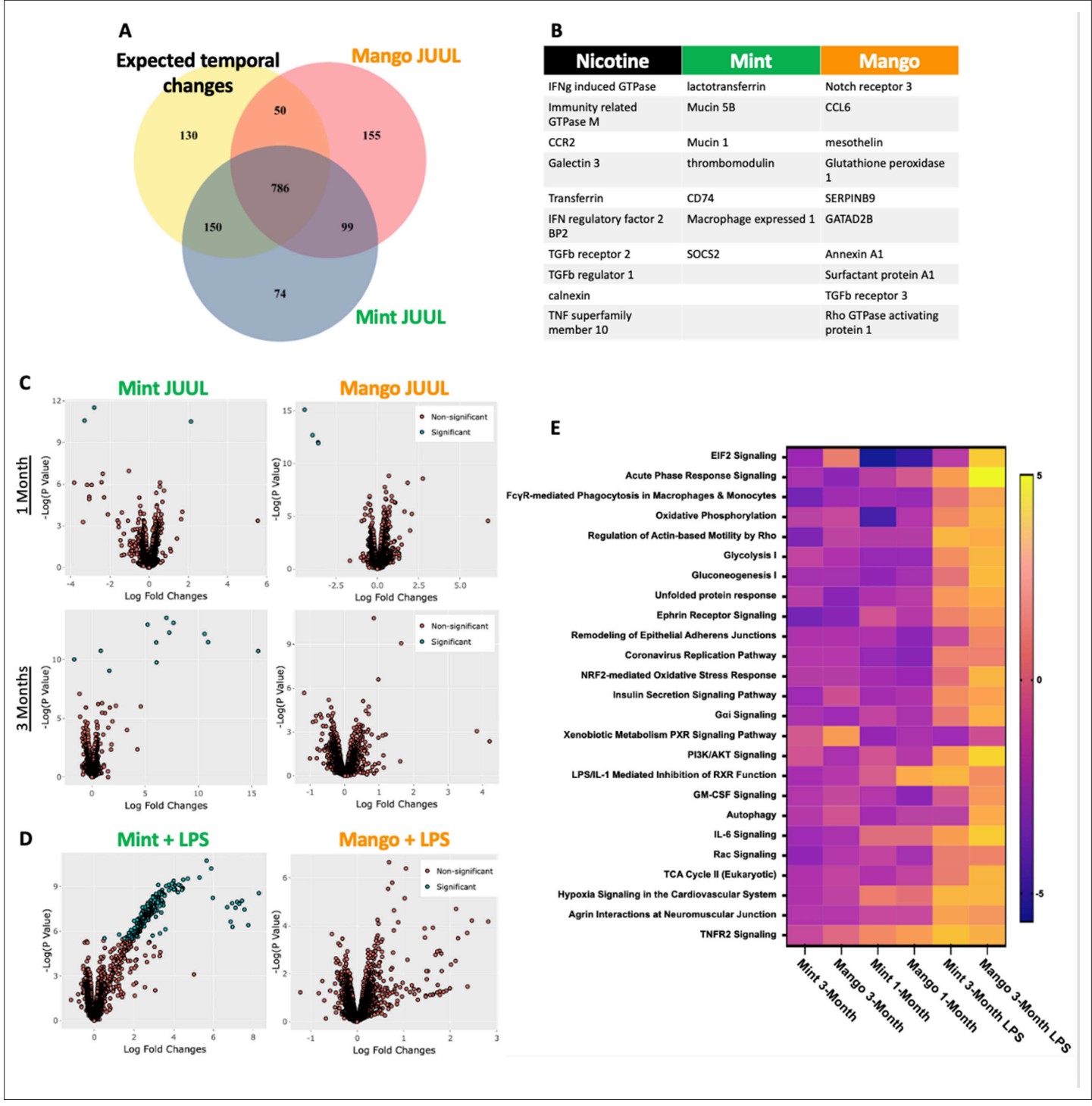

**Figure 6.** Unique RNAseq signatures in the lungs exposed to different flavors of JUUL aerosols. (**A**) Venn diagram of gene expression unique to JUUL Mango (155) and JUUL Mint (*Alhaddad et al., 2014b*). Gene expression changes common to both aerosols (99) suggest that they are due to chemicals found in both of the flavored e-liquids (nicotinic salts, propylene glycol, glycerin, benzoic acid, etc). (**B**) Greatest gene expression changes associated with nicotine and flavorant chemicals within aerosols. (**C**) Volcano plots demonstrating gene expression changes specific for each flavor after sub-acute exposure (1 month; top row) and 3 months of exposure (bottom row). (**D**) Gene expression changes associated with inhalation of JUUL aerosols with different flavors in the setting of inflammatory challenge with inhaled LPS.(**E**) Heat map of the pathways most notably impacted by 1- and 3-month daily exposures to JUUL Mint and Mango, as well as in the setting of LPS challenge at 3 months. IFNg: interferon gamma; CCR: C-C motif chemokine related; IFN: interferon; BP: binding protein; TGFb: transforming growth factor beta; TNF: tumor necrosis factor; SOCS: suppressor of cytokine signaling; CCL: C-C motif ligand.

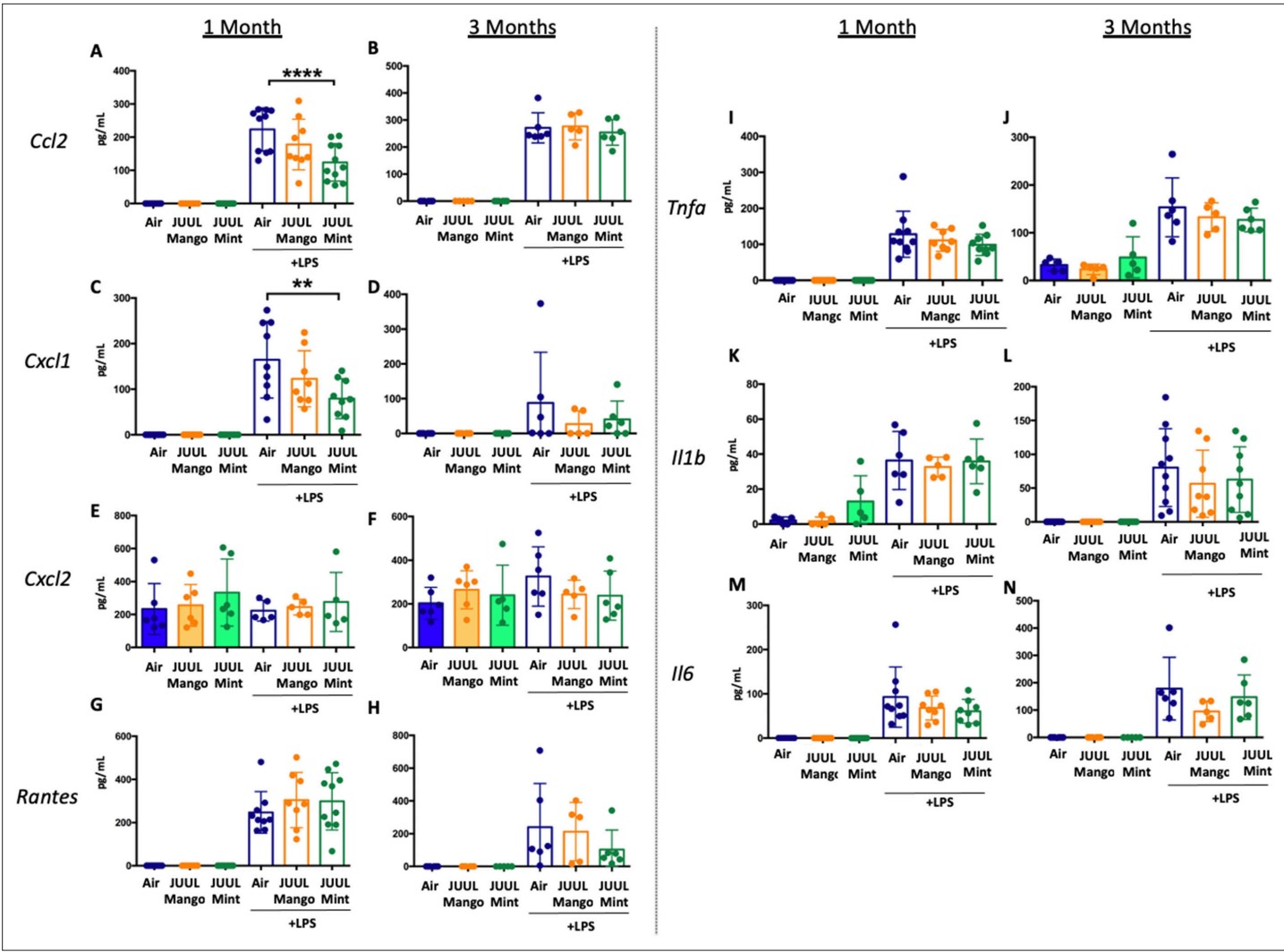

**Figure 7.** JUUL exposure alters airway inflammatory responses in the setting of inhaled LPS challenge. BAL was harvested at the endpoints, and cytokines and chemokines were quantified by ELISA. *Ccl2* at (**A**) 1 month, and (**B**) 3 months, *Cxcl1* at (**C**) 1 month and (**D**) 3 months, *Cxcl2* at (**E**) 1 month and (**F**) 3 month, RANTES at (**G**) 1 month and (**H**) 3 months, *Tnfa* at (**I**) 1 month and (**J**) 3 months, *Il1b* at (**K**) 1 month and (**L**) 3 months, *Il6* at (**M**) 1 month and (**N**) 3 months. Data are presented as individual data points ± SEM with n = 5–11 mice per group. **p < 0.01 and **** p < 0.0001.

The online version of this article includes the following figure supplement(s) for figure 7:

**Figure supplement 1.** Inflammatory cytokine levels in the BAL of mice exposed to JUUL Mango and JUUL Mint prior to inhaled LPS challenge.

vaping on the severity of attendant respiratory diseases (*Madison et al., 2019*). Under homeostatic conditions, the BAL of JUUL Mango and JUUL Mint mice contains similar levels of inflammatory cytokines and chemokines at both 1 and 3 months (*Figure 7A–N*). Inhaled LPS is a model of Gram-negative bacterial pneumonia and acute lung injury in mice. Mice exposed to JUUL aerosols and challenged with inhaled LPS had similar total numbers of leukocytes and neutrophils in the airways relative to Air controls (*Appendix 1—figure 3*), and histological analysis of H&E staining showed that parenchymal inflammation was similar across groups after LPS challenge (*Appendix 1—figure 3*). LPS challenge also leads to increased levels of *Ccl2* and *Kc/Cxcl1* murine homolog of *Il8* in the airways. The increases in *Ccl2* and *Cxcl1* elicited by LPS were diminished in mice exposed to JUUL Mint, demonstrating an attenuated inflammatory response to LPS after sub-acute exposure to JUUL (*Figure 7A and C,* respectively). Differences in LPS induced cytokine levels were no longer observed after 3-month JUUL exposure versus Air control groups (*Figure 7B, D, F, H, J, L and N*; *Figure 7—figure supplement 1*), suggesting that chronic use of JUUL may not alter inflammatory responses to Gram-negative infections in the lung.

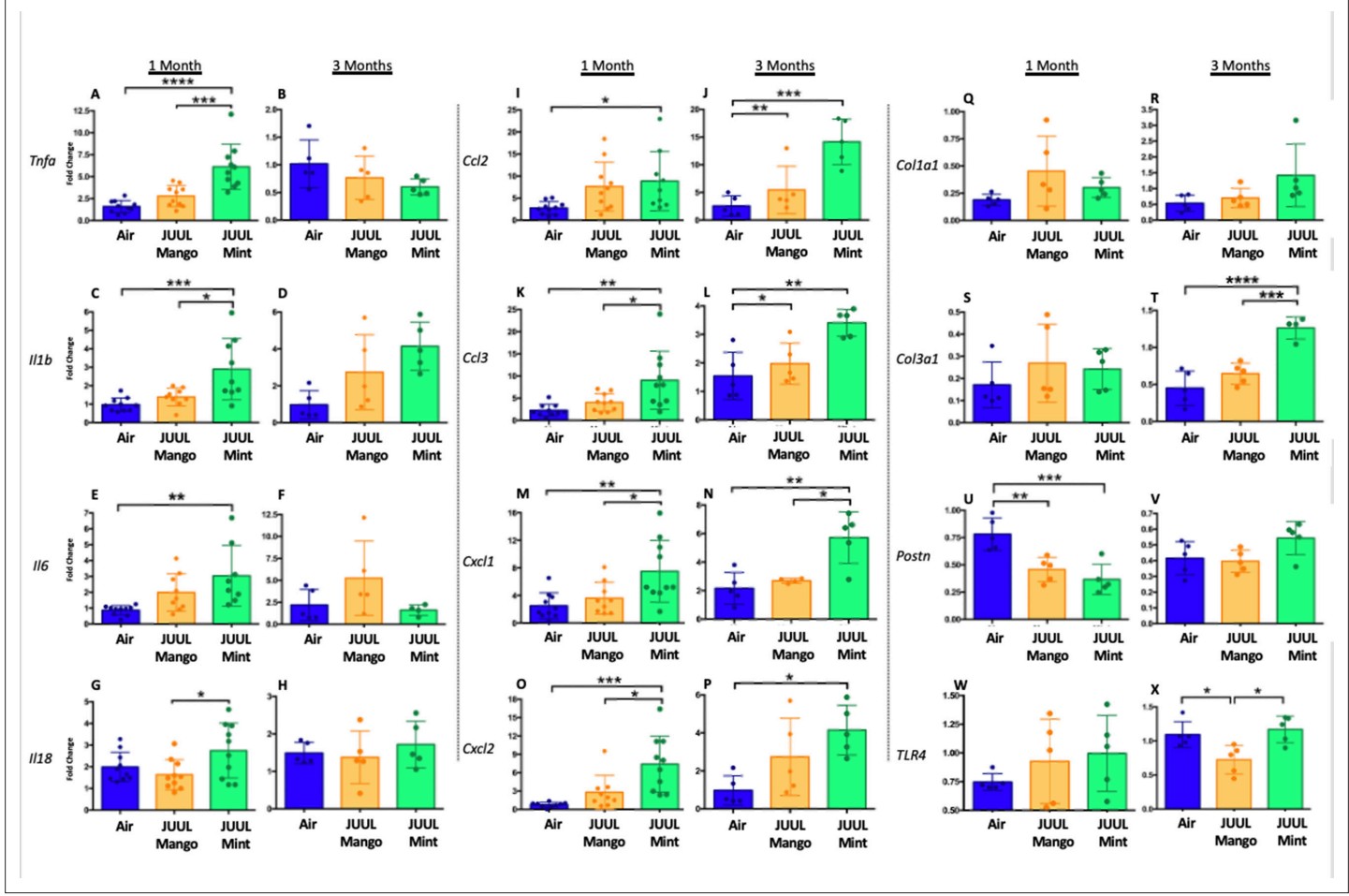

**Figure 8.** Cardiac inflammation induced by inhaled LPS challenge is increased in the setting of 3 months of JUUL aerosol inhalation. Hearts were harvested, and RNA was extracted from the left ventricle and qPCR was performed to quantify the gene expression of different cytokines, chemokines and fibrosis-associated genes. Cytokines include *Tnfa* at (**A**) 1 month and (**B**) 3 months, *Il1b* at (**C**) 1 month and (**D**) 3 months, *Il6* at (**E**) 1 month and (**F**) 3 months, and *Il18* at (**G**) 1 month and (**H**) 3 months. Chemokines include *Ccl2* at (**I**) 1 month and (**J**) 3 months, *Ccl3* at (**K**) 1 month and (**L**) 3 months, *Cxcl1* at (**M**) 1 month and (**N**) 3 months, and *Cxcl2* at (**O**) 1 month and (**P**) 3 months. Fibrosis-associated genes include *Col1a1* at (**Q**) 1 month and (**R**) 3 months, *Col3a1* at (**S**) 1 month and (**T**) 3 months, *Postn* at (**U**) 1 month and (**V**) 3 months, and *Tlr4* at (**W**) 1 month and (**X**) 3 months. Changes in expression levels are relative to Air controls. Data are presented as individual data points ± SEM with n = 5–11 mice per group. *p < 0.05, **p < 0.01, ***p < 0.001 and ****p < 0.0001.

The online version of this article includes the following figure supplement(s) for figure 8:

**Figure supplement 1.** Inflammatory gene expression changes in the hearts of mice exposed to JUUL Mango and JUUL Mint and challenged with inhaled LPS.

## Cardiac inflammation induced by inhaled LPS challenge is increased in the setting of JUUL aerosol inhalation for 3 months

Bacterial pneumonia and acute lung injury lead to inflammation not only in the lungs and systemic circulation, but also in the heart (*Morris, 2014*). It is common for patients to develop myocardial inflammation and even ischemia during lung infections (*Morris, 2014*; *Feldman et al., 2019*). Tobacco smoking is well known to increase cardiovascular diseases and worsen outcomes in the setting of pneumonia (*Crotty Alexander et al., 2015a*; *Kondo et al., 2019*) and recently, it has been suggested that dual use of e-cigarettes with conventional tobacco leads to significantly higher odds of cardio-vascular disease compared with cigarette smoking alone (*Osei et al., 2019*). Thus, we assessed the impact of acute lung injury on inflammation in cardiac tissues of JUUL exposed mice.

We assessed expression of *Tnfa, Il1bIl6, Il18, Ccl2, Ccl3, Cxcl1, Cxcl2, Col1a1, Col3a1, Postn*, and *Tlr4* at 1 and 3 months to determine if the LPS challenge caused changes in cardiac inflammation in

the setting of JUUL aerosol inhalation (*Figure 8A–X*). LPS challenge of mice exposed to JUUL Mint for 1 month led to significantly greater expression of cytokines (*Tnfa, Il1b, Il6*) and chemokines (*Ccl2, Ccl3, Cxcl1, Cxcl2*) than observed in Air controls (*Figure 8A, C, E, I, K, M and O*). The enhanced chemokine induction was further elevated after 3 months of exposure to JUUL Mint (*Figure 8J, L, N and P*). In contrast to the elevated inflammatory response to LPS observed in mice exposed to JUUL Mint, JUUL Mango exposed mice did not have enhanced expression of cytokines or chemokines. Indeed, the effects of 1 month JUUL Mint versus JUUL Mango exposure were statistically significant with regard to changes in *Tnfa, Il1b, Il18, Ccl3, Cxcl2* (*Figure 8A, B, G, K and M*) as well as on *Cxcl1* expression at both 1 and 3 months JUUL exposure (*Figure 8M–N*).

Enhanced inflammatory responses within tissues are known to result in fibrosis in some cases. However, analysis of pro-fibrotic gene expressiononly revealed increased *Col3a1* after 3 months of JUUL Mint exposure (*Figure 8T*). *Col1a1* and *Tlr4* expression were not higher in the JUUL exposed groups. Indeed, periostin expression was lower in 1-month JUUL Mint and JUUL Mango compared to Air (*Figure 8U*) and *Tlr4* expression was also lower in the 3-month JUUL Mango group (*Figure 8X*; *Figure 8—figure supplement 1*). Thus, while fibrotic changes are not evident, the enhanced expression of chemokines and cytokines in the presence of cardiac stress indicates that the use of JUUL devices could predispose to cardiac tissue damage by exacerbated inflammation. In addition, we consistently found more profound effects of JUUL Mint on inflammatory cytokine and chemokine

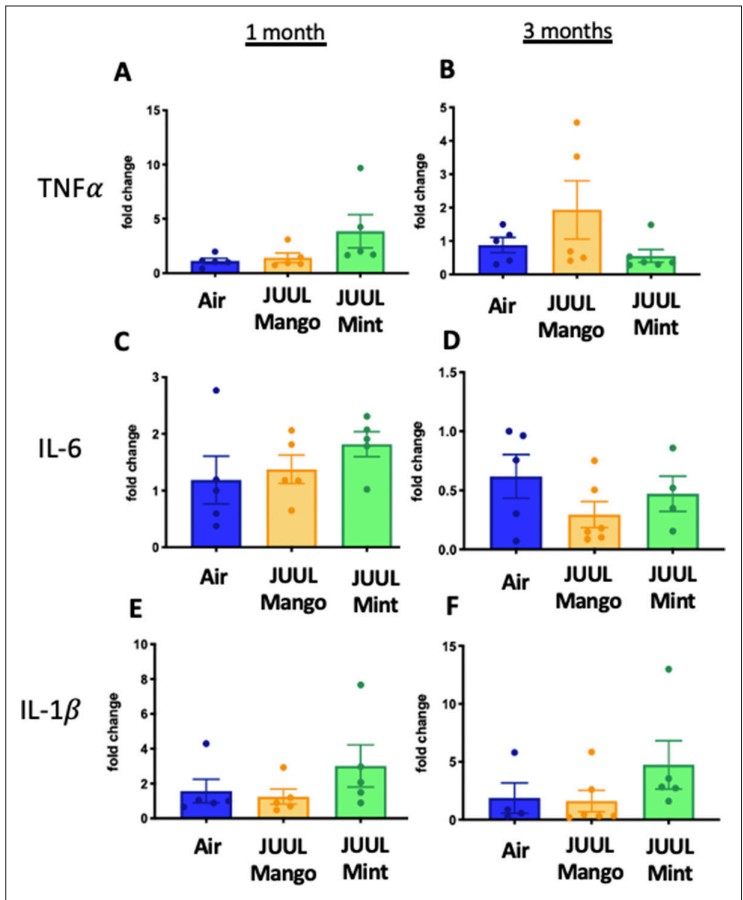

**Figure 9.** Three months of JUUL aerosol inhalation does not alter inflammatory markers in the setting of by inhaled LPS challengein the gastrointestinal tract. Inflammation was assessed in the colon at 1 and 3 months. Panels show inflammation markers in the colon in *Tnf* (**A**) 1 month and (**B**) 3 months, *Il6* at (**C**) 1 month and (**D**) 3 months, *Il1b* at (**E**) 1 month and (**F**) 3 months. Data for inflammation markers are presented as individual data points ± SEM.

The online version of this article includes the following figure supplement(s) for figure 9:

**Figure supplement 1.** Inflammatory gene expression in the colon of mice exposed to JUUL Mango and JUUL Mint and challenged with inhaled LPS.

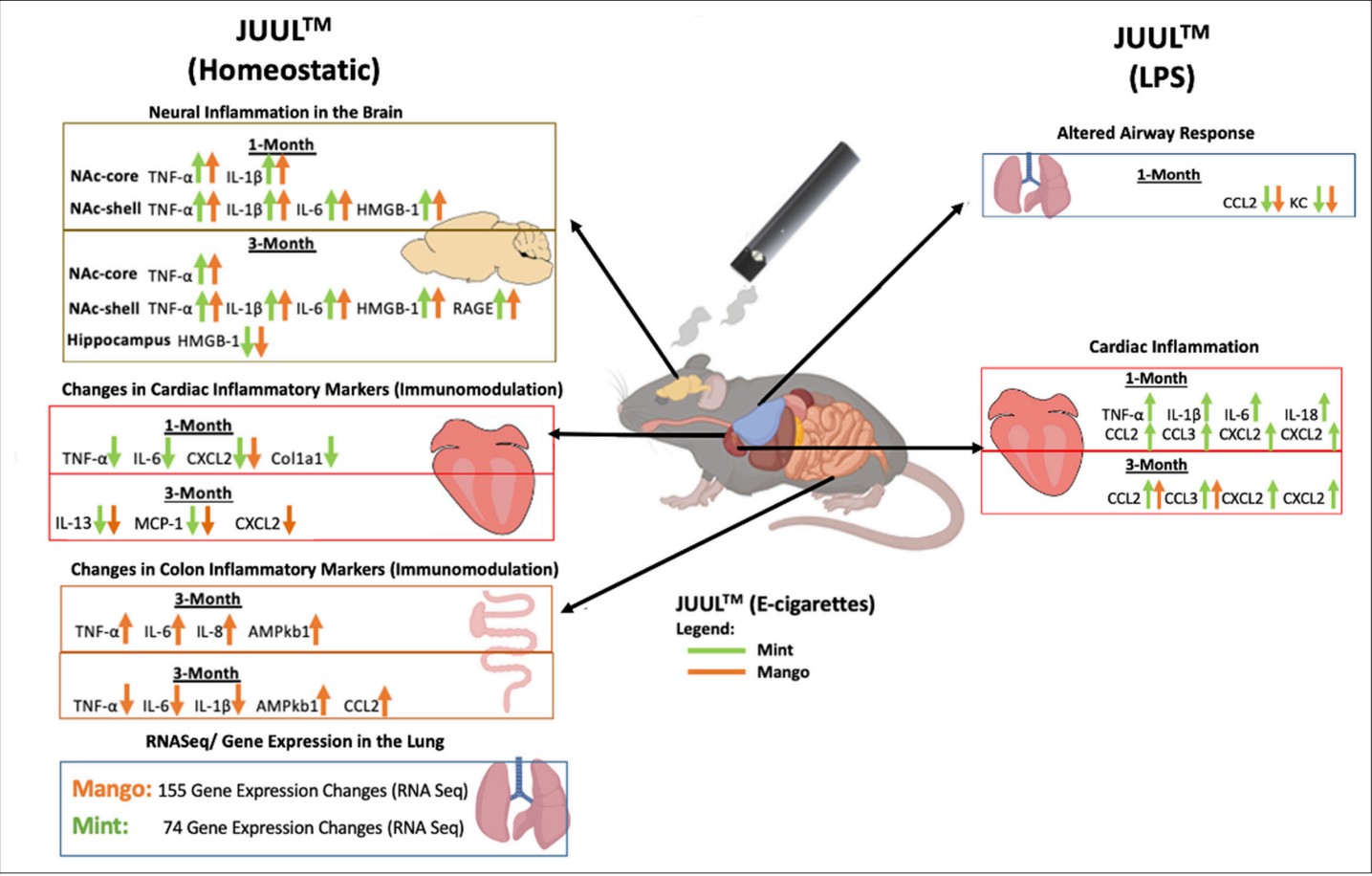

**Figure 10.** Overview of JUUL aerosol induced inflammatory changes across organs.

gene expression, suggesting that components of the e-cigarette flavoring play a significant role in heart inflammation.

## JUUL aerosol inhalation for 3 months followed by inhaled LPS challenge does not alter protein markers of the gastrointestinal tract

We showedthat JUUL exposure affected the expression of pro-inflammatory cytokines in colonic tissue under homeostatic conditions. Hence, we also assessed whether this effect would be exacerbated in the context of inhaled LPS challenge. No greater increases in *Tnfa*, *Il6*, or *Il1b* following LPS treatment were observed in mice subjected to 1 of 3 months of JUUL exposure (*Figure 9*; *Figure 9—figure supplement 1*).

## Discussion

E-cigarette use has been linked to adverse cardiovascular (*Lee et al., 2018*) and immune responses (*Corriden et al., 2020*; *Hwang et al., 2016*). However, little is known about the effects of e-cigarette use on the brain and gastrointestinal system. In this study, we found that mice exposed to flavored JUUL aerosols for 1 and 3 monthshad significant neuroinflammation in the brain, as well as inflammatory modulation in the heart, lung and GI tract (*Figure 10*).However, we did not detect changes in cardiopulmonary physiology with daily inhalation of JUUL aerosols for 1 month (generally considered a subacute duration of exposure) or 3 months (generally considered a chronic exposure in mice).While many of the inflammatory and immunomodulatory findings were solely at the gene expression level, they do support our hypothesis that daily inhalation of flavored, high-nicotine (59 mg/mL) e-cigarette aerosols induces immunomodulation broadly across organ systems. These immunomodulatory

changes caused by e-cig use are concerning for potential down-stream effects at both the organ-level and on overall health.

## Central nervous system

The nucleus accumbens is part of the reward pathways in the central nervous system and is critical in regulating motivation, reward and addiction behaviors. The nucleus accumbens (NAc) in particular was found to have elevated levels of inflammatory markers in response to daily inhalation of JUUL aerosols, including *Tnfa*, *Il1b,* and *Il6* in both NAc-core and NAc-shell, and HMGB1-1 and RAGE in the NAc-shell (*Heldt et al., 2020*). This evidence of inflammation is concerning as the NAc-core and NAc-shellare both known to contribute to the formation of anxious or depressive behaviors in the context of neuroinflammation via NFκB signaling pathway (*Mavridis, 2015*; *Décarie-Spain et al., 2018*).

The NAc-core and NAc-shell are also known to control reward-related behaviors through distinct neurocircuitry (*Augur et al., 2016*; *Keistler et al., 2015*; *Stefanik et al., 2016*). We have previously shown that chronic exposure to ethanol is associated with dysregulation of the glutamatergic system and neuroinflammatory response in the NAc-shell but not in the NAc-core (*Alhaddad et al., 2020*; *Alasmari et al., 2020*). Recently, we found that 3-month exposure to either JUUL Mint or Mango may have contributed to dysregulation in glutamatergic system in the NAc-shell (*Bennett et al., 2018*). In general, the new neuroinflammatory data we present here further suggests the deleterious effects of JUUL exposure on the nucleus accumbens. The hippocampus, which is essential for learning and memory (*Sharma et al., 2020*),showedsignificant changes in HMGB1. This is concerning because increased HMGB1 expression has been found to be a marker of neuroinflammatory conditions and may be a predictor of cognitive decline (*Paudel et al., 2018*).

Exposure to drugs such as methamphetamine, cocaine and ethanol activates neuroinflammatory pathways thatare associated withthe release of HMGB1-1 in the striatum and nucleus accumbens, as part of addictive behaviors and drug reward (*Alasmari et al., 2020*; *Gao et al., 2020*). The overall similarity in inflammatory profiles and brain regions between drugs of abuse and that observed in this model of chronic JUUL exposure (inhalation for 3 months) raises concern as it suggests that e-cigarette use is associated with addictive behaviors (*Vogel et al., 2020*). Further studies into the overlap of induced neuroinflammatory pathways between drugs of abuse and JUUL is required to better understand these relationships. This includes studies in human subjects to assess the incidence of anxiety and depression in JUUL users. Indeed, neuroinflammatory effects caused by chronic, daily JUUL exposure may lead to adaptations in neural circuitry that promote addictive behaviors and drug dependence, providing a neurophysiological explanation for the observation that e-cigarette use does not help with smoking cessation (*Alasmari et al., 2017*).

It is well established that high concentrations of nicotine inhalation are toxic to the human body in a variety of ways (*Mishra et al., 2015*). JUUL pods have been found to have the highest nicotine concentration (up to 10 times more) of any of the other cartomizer style e-cigarette or refill fluids (*Omaiye et al., 2019*). Previous studies utilizing continuous systemic delivery of nicotine via implanted pumps concluded that nicotine did not contribute to the development of neuroinflammation. The differences in findings between this study and previous work could be explained by differences in the mode of nicotine delivery, type of nicotine (free-base nicotine and nicotinic salts), inhalant device or other chemicals contained in the vaping aerosols, including vehicle components. Notably, because Mint and Mango effects differed, several of our findings point to a 'non-nicotine' chemical flavorant component of the JUUL device that may be driving inflammatory changes in the brain. Recent study into the effects of vaping on the blood brain barrier lends further support to this theory, as pro-inflammatory changes were observed, partly independent of nicotine content (*Heldt et al., 2020*).

## Pulmonary

As the entry point to the body for inhalants, the lungs get a high level of exposure to the chemicals within e-cigarette aerosols (*Grana et al., 2014*). Because the size of e-cigarette aerosols, they reach deep into the lungs, even further than conventional tobacco smoke (*Manigrasso et al., 2015*). While it may take decades for us to fully understand the pathologic effects of e-cigarette aerosols on the lungs, animal models can provide some insight as to what physiologic, inflammatory, and even carcinogenic effects may occur. While prior studies using earlier generations of e-devices and e-liquids have found numerous changes in the lungs, we found that 1-month and 3-month JUUL Mint and

Mango exposures did not induce airway hyperreactivity, emphysematous changes, or overt inflammation. However, the gene expression changes detected within the lung parenchyma raise concern for an altered phenotypic state. These changes are most ominous for their likelihood of altering the lungs responses to challenges, such as bacteria, viruses, smoke and pollution. The association of JUUL Mint exposure with a unique set of gene expression changes in the setting of inhaled LPS challenge further supports this theory, demonstrating that e-cigarette exposed lung tissue responds differently to an inflammatory challenge. And as we have learned time and time again in the immune and inflammatory fields, any tipping of the balance will lead to pathology and disease.

## E-cigarette flavors

While the nicotine concentration in JUUL pods is quite high, it does not vary with the JUUL flavor (*Omaiye et al., 2019*). The basis for the variation between the two different JUUL flavored pods tested in our study is most likely due to the differences in chemical flavorants. We observed significantly different inflammatory gene alterations in cardiac and colonic tissue in response to exposure to Mint and Mango JUUL aerosols for 3 months (*Figure 9*). Similarly, Omaiye et al reported a variety of different flavoring chemical across JUUL flavors and demonstrated that aerosols generated from different JUUL flavors induced different levels of cytotoxicity in BEAS-2B cells in vitro, in a flavor-dependent manner (*Omaiye et al., 2019*). Two specific chemicals found in multiple flavors are ethyl maltol and menthol. Ethyl maltol concentrations have been shown to be highest in Mango pods (1 mg/ml), while menthol concentrations are highest in cool Mint pods (10 mg/ml) (*Omaiye et al., 2019*). The most remarkable variations we observed were in response to acute lung injury through LPS challenge, where significantly higher levels of cardiac inflammatory genes were seen in mice exposed to Mint relative to Mango and controls. In the brain, inhalation of JUUL Mint aerosols led to higher *Tnfa* and *Il1b* in the NAc-shell relative to JUUL Mango. Mint aerosols are highly similar to menthol aerosols and previous studies have shown greater increases in neuronal nAChR receptors after exposure to nicotine with menthol relative to nicotine exposure alone (*Henderson et al., 2017*). As a result, we surmise that the flavoring compound menthol in 'Cool Mint' could be a factor in the differences seen in the effects of Mint vs Mango. Overall, these findings suggest that components other than nicotine may contribute to the observed neuroinflammatory changes. Further research is needed to better understand how specific, non-nicotine JUUL components contribute to inflammatory and neuronal effects.

## Fibrosis

Collagen expression is a hallmark of fibrosis and has previously been observed in studies involving combustible cigarette smoke (*Drummond et al., 2016*). However, compared to our prior study with Vape pens (using the nose-only InExpose system by SciReq) where profound increases in collagen deposition were observed across cardiac, hepatic and renal issues (*Crotty Alexander et al., 2018*), we did not find increased fibrosis in these same organs in JUUL exposed mice. This raises questions about the role of different e-cigarette devices, e-liquids and experimental approaches for aerosol exposures and suggests differences in the chemical composition of aerosol and its delivery as potential causes of different biological outcomes. Research in this area is thus complex, as many teams are using a variety of devices and liquids, which may produce different effects on mammalian systems.

## Colon

Intensive research has been done on the effect of cigarette smoke on inflammation and the pathogenesis of diseases such as Ulcerative Colitis and Crohn's Disease, however there are mixed, somewhat inconclusive results when it comes to whether this exposure leads to long-term activation or suppression of inflammatory pathways, and their relation to the likelihood of developing these gastrointestinal afflictions (*Verschuere et al., 2012*). Nicotine specifically has been previously found to decrease the expression of pro-inflammatory cytokines in the colon (*Van Dijk et al., 1995*). Here, we saw an increase in pro-inflammatory cytokines in the colon after 1 month of exposure, whereas these same signals were significantly decreased after 3 months of exposure when compared to the control group (*Figure 9*). Thus, over the course of the 3 months, the body may adapt to these changes and downregulate these markers significantly through some yet unidentified mechanism, pathway, or interaction with the specific components in the JUUL device. Whether this inflammatory adaption is beneficial or detrimental to the overall health of the colon remains to be defined.

## Cardiac

Bacterial pneumonia and acute lung injury are known to cause inflammation systemically and in the heart (*Morris, 2014*). Indeed, effects of viral infections such as SARS-CoV-2, while originating in the lung, appear to also signal to the heart, where pathological inflammation and cardiac dysfunction are observed (*Lindner et al., 2020*; *Nishiga et al., 2020*). The importance of cardiac inflammation in development of heart failure following viral infection, myocardial infarction and non-ischemic cardiac injury has been increasingly appreciated (*Adamo et al., 2020*). For example, *Il1b* blockade has been shown to diminish adverse cardiac events and heart failure progression (*Everett et al., 2019*; *Abbate et al., 2020*). We demonstrate here that the hearts of mice subject to chronic JUUL exposure are more sensitive to the effects of LPS delivered to the lung than are Air control mice, as evidenced by enhanced expression of pro-inflammatory cytokines and chemokines including *Il1b*. The observation that there were no significant changes in vagal tone (assessed by heart rate variability) and that the pro-inflammatory enhancement by JUUL exposure was largely confined to JUUL Mint, suggests that this is not due to signals generated by direct nicotine action. While the mechanism by which chronic JUUL exposure predisposes to LPS-induced cardiac inflammation remains to be determined, these findings suggest that chronic JUUL inhalation could lead to systemic changes which sensitize maladaptive inflammatory responses that affect cardiac function.

## Limitations

Contrary to our initial expectations, we did not find significant changes in autonomic tone or pulmonary function with daily, long-term JUUL aerosol exposure. Our model is limited in that mice are primarily nose breathers and we used whole-body exposure, so it is possible that the extent of e-cigarette aerosol exposure at the level of the alveoli may be lower than in humans due to aerosol deposition within the nasal cavity. Alternatively, our study may be underpowered to detect subtle differences induced by JUUL vaping. Furthermore, our study only assessed the impact of daily JUUL aerosol inhalation for 1 and 3 month-long exposures and thus cannot ensure that other effects may be seen with more chronic, 6–12 month, exposures.

Our study also used a 5 day per week exposure model, which is common practice in many exposure labs due to the difficulty and onerous nature of 7-day exposure models for months on end, but is a limitation in that the 2 day breaks over weekends may have allowed for inflammatory and physiologic recovery. These studies were conducted in female mice, such that findings may not be applicable in males. While humans use e-cigarettes throughout the day, pulling out their e-device and vaping numerous times, this use pattern was not accurately mimicked in this study where the exposures were run three times daily. However, this limit to 20 min three times daily was necessary to allow mice access to food and water, and to keep their stress level minimal.A limitation to any future studies is that the Mint and Mango JUUL flavors can no longer be studied since they have been discontinued. While these two flavors are no longer available they were composed of chemicals commonly found in other flavors, such that these data are likely relevant to other vapes. In particular, JUUL Mint shares chemical features with JUUL Menthol, which took its place as one of the most popular JUUL flavors.

It is important to mention that this is the first study to assess JUUL devices and flavorants in a multiorgan fashion. We found effects of JUUL aerosols on inflammatory responses in organs other than the lungs, demonstrating that the effects of e-cigarette exposure may be far-reaching across the body, as are those of cigarette smoke. While this study identified inflammation within brain regions, raising concern for both behavioral and psychological effects in human vapers, dedicated studies of the effects of e-cigarette use on behavior and mood states are needed. While the majority of changes in cytokine levels induced by e-cigarette exposure across organ systems were relatively small, the fact that multiple cytokines changed in concert indicates a significant shift in immunophenotyping across organs. We are most concerned about how these shifts in the inflammatory state will alter an e-cigarette users response to common clinical challenges. For example, Madison et al. exposed mice to e-cigarette aerosols with and without nicotine and found that both exposures increased susceptibility to acute lung injury in the setting of viral pneumonia (*Madison et al., 2019*). In our work, we utilized the LPS model of acute lung injury to take a first look at the potential impact of JUUL inhalation in particular on susceptibility to lung inflammation. Further work is needed to truly define how the subtle, broad shifts in the cytokine milieu across organs will impact the health of e-cigarette vapers.

## Conclusion

Our findings suggest that chronic inhalation of chemicals within e-cigarette aerosols can lead to inflammatory changes across multiple organ systems. JUUL users may unwittingly expose themselves to increased neurologic, colonic and cardiac risk. Further research isgreatlyneeded to better understand the long-lasting effects of vaping.

# Materials and methods

## JUUL exposures

Six to eight week old female C57BL/6 mice were purchased from Envigo. After 2 days to allow mice to equilibrate with their environment, all mice were placed into a large enclosure to allow mixing of the microbiome for one hour. Mice were placed three per cage and each cage was randomly assigned to the type of exposure ( ≥ 2 cages per exposure). Mice were placed in individual sections of a full-body exposure chamber (Scireq) for 20 min three times daily, for a total of 60 min per day, for 4–12 weeks. Mice were exposed to either e-cigarette aerosol created from Mango JUUL pods or Mint JUUL pods containing 5% nicotinic salts (59 mg/ml) using the InExpose system (Scireq). Air control mice were placed in an identical chamber for the same amount of time but inhaled room air only. A 3-D printed adapter was created to produce a tight fit for the JUUL device (designed and produced by Vitorino Scientific LLC). A negative pressure of 2 L/s was used to activate the e-cigarette for 4 s followed by 16 s of room air at 2 L/s. The final exposure was done 30 min prior to harvest. All experiments were conducted with approval of the UCSD Institutional Animal Care and Use Committee (IACUC protocol S16021). All authors complied with the ARRIVE guidelines.

## LPS intranasal challenge

Mice were sedated with isoflurane, held upright and intranasally challenged with LPS (*E. coli* O111:B4; Sigma) at a concentration of 2.5 µg per gram of mouse in 0.9% saline (100 µl). The LPS challenge was given through the left nare to decrease liquid trapping in the nasopharyngeal dead space. Mice were maintained in the upright position until respirations returned to normal. Mice were monitored overnight prior to harvest 24 hr after challenge.

## Assessment of pulmonary function

At the end of 4 and 12 weeks of exposure, prior to harvest, mice were sedated via intraperitoneal (i.p.) injection of ketamine 10 mg/ml xylazine 100 mg/ml. Mice underwent tracheostomy with 18 g metal cannula and were attached to the FlexiVent mouse ventilator (SciReq). Measurements of lung physiology via mechanic scans were obtained at baseline, followed by assessment of physiologic responses to methacholine (MCH) challenge at 0, 6, 12, and 24 mg/ml, including Respiratory Resistance (Rrs) and Elastance (Ers). Pressure Volume (PV) loops were also obtained. Surgeons and operators of the FlexiVent were blinded to the exposure group, and mice were harvested in an alternating pattern across groups (e.g. JUUL Mint, Air, JUUL Mango, JUUL Mint, JUUL Mango, Air, etc.).

## Cell counts and differential

Bronchoalveolar lavage (BAL) was collected by flushing airways with ice cold 800 µl PBS three times via mouse tracheal cannulation. Samples were pelleted at 3000 rpm for 4 min at 4 °C. Pellets were resuspended in 1 ml of ice cold PBS, counted with Countess (Life Technologies) for total cells quantification. Two dilutions (1:1 and 1:4) of total cells were cytospun onto slides at 800 rpm for 3 min and then cells were fixed with Giemsa Wright. Slides were de-identified and randomized prior to blinded cell counting; 200 cells from each slide were counted via light microscopy under 40 X magnification, and percentage of neutrophils was calculated and total amounts of neutrophils extrapolated based on total cell quantification.

## Histology and mean linear intercept quantification

Lungs were inflated with Zfix (Anatech ltd) at 25 cm $H_2O$ pressure for 18 hr, followed by transfer into 75% ethanol prior to paraffin embedding. Lung slices were stained with H&E. Mean Linear Intercept (MLI) was assessed by sampling a single sagittal tissue section containing large, medium and small airways from each lung specimen. The lung sections were digitally scanned at ×10 magnification,

yielding an effective isotropicpixel size of 0.441 µm. The digitally scanned lung sections were analyzed using FIJI ImageJ. MLI was estimated using the intersection counting method. A two-dimensional square grid, of side length 200 µm, was superimposed on the lung image at full resolution. A random offset was applied to the grid independently for each image and grid fields that did not fall completely within the lung parenchymawere excluded from analysis. A subset of the eligible grid fields (~30 per lung) was chosen using the principles of systemic uniform random sampling (SURS) to optimize unbiased sampling. The total number of lung fields in the section were counted from the top left edge of the image, progressing back and forth until all sections were identified. Every $n^{th}$ field was chosen for analysis where $n$ equals the nearest divisor of the total count divided by 30. The first field for analysis was chosen at random from the first $n$ fields. The total number of intersections of the test lines with alveoli and alveolar ducts were estimated manually, eliminating small airways and vessels. The proportion of end points falling on septal tissue and within airspaces were also estimated. MLI was calculated using the formula:

$$Lm = 2.d.\frac{P}{I}$$

where d is the length of the test line (200 µm), $P$ is the proportion of points falling in airspaces, and $I$ is the number of total intersections of the test line with alveolar airspaces.Statistical comparison of the MLI between groups was performed using two-way analysis of variance (two-way ANOVA).

## Fibrosis analysis

The right kidney, one lobe of liver, and the base of the heart were then immediately dissected after euthanasia and placed in Z-fix at 4 °C. After 48 hr, all organs were moved to 75% ethanol and submitted to the University of California, San Diego histology core for paraffin embedding. Collagen was detected in 5 µm sections first by Masson's trichrome stain. All histology slides underwent quantification of fibrosis by calculating the mean percent fibrotic area in 15–25 randomly acquired ~20 images using computer-aided morphometry performed using ImageJ. Briefly, using the color threshold with default thresholding method, red threshold color and HSB color space, the total area of tissue in the slide was selected and measured, later the tissue stained for Masson's trichrome blue was also selected and measured prior adjustment of the 'Hue' parameter (Saturation Brightness/Value Each color shade). Then, a percentage of the area stained by Masson's trichrome blue was determined relative to the total tissue area. All histology slides from the same tissue group were blinded and underwent these computer analyses in an identical fashion. Fibrotic area is presented relative to that of air controls.

## Isolation of RNA from the murine colonic tissue and qRT-PCR for inflammatory cytokines

RNA was isolated from mouse colon tissues using the Zymo miniprep kit according to the manufacturer's instructions, followed by cDNA synthesis. Quantitative Real-Time PCR was conducted for target genes and normalized to housekeeping gene 18 S rRNA. Primer sequences are provided in *Table 1*.

## Cardiovascular physiology measurements

Heart rate, heart rate variability (HRV) and blood pressure measurements were taken after the last exposure to JUUL aerosol or Air at 1 and 3 months, via the Emka non-invasive ECG Tunnels and the

**Table 1.** Primer sequences for qRT-PCR on colonic tissues.

| qPCR primers (Mouse) | Forward primer (3'- 5') | Reverse primer (3'- 5') |
| --- | --- | --- |
| Mouse 18 s | GTAACCCGTTGAACCCCATT | CCATCCAATCGGTAGTAGCG |
| Mouse Il6 | CCCCAATTTCCAATGCTCTC C | CGCACTAGGTTTGCCGAGTA |
| Mouse Il1b | GAAATGCCACCTTTTGACAG T | CTGGATGCTCTCATCAGGAC A |
| Mouse Tnfa | CCACCACGCTCTTCTGTCTA | AGGGTCTGGGCCATAGAAC T |
| Mouse Il8 | CCTGCTCTGTCACCGATG | CAGGGCAAAGAACAGGTCA G |
| Mouse Ccl2 | AAGTGCAGAGAGCCAGACG | TCAGTGAGAGTTGGCTGGTG |

**Table 2.** Primer sequences for qRT-PCR on brain tissues.

| Targets | Primers | Sequences | References |
|---|---|---|---|
| Gapdh | Forward (Sense) | 5'-ATGACATCAAGAAGGTGGTG-3' | *Sandhir et al., 2008* |
| | Reverse (Antisense) | 5'-CATACCAGGAAATGAGSCTTG-3' | |
| Il1b | Forward (Sense) | CCAGCTTCAAATCTCACAGCAG | *Kawane et al., 2010* |
| | Reverse (Antisense) | CTTCTTTGGGTATTGCTTGGGATC | |
| Tnfa | Forward (Sense) | CACAGAAAGCATGATCCGCGACGT | *Kawane et al., 2010* |
| | Reverse (Antisense) | CGGCAGAGAGGAGGTTGACTTTCT | |
| Il6 | Forward (Sense) | TCCAGTTGCCTTCTTGGGAC | *Kawane et al., 2010* |
| | Reverse (Antisense) | GTACTCCAGAAGACCAGAGG | |

CODA non-invasive blood pressure system. Prior to data collection, mice were acclimated for 20 min per day for the last 3 days in the ECG and blood pressure systems. Heart rate variability was determined through time-domain measurements, specifically SDNN and RMSSD. The SDNN is the standard deviation of all normal R-R intervals, providing information on total autonomic variability. The RMSSD is the root mean square of those standard deviations and represents the variability in the short term.

## Brain tissue harvesting

At the end of the experiments, mice were euthanized by ketamine and xylazine i.p. injection, rapidly decapitated, with their brains removed and stored at –80 °C. The cryostat apparatus maintained at –20 °C and used to dissect NAc-core, NAc-shell, and HIP, which micropunched stereotaxically. The stereotaxic coordinates for the mice brain (*Paxinos et al., 2020*) was used to isolate the brain regions of interest following visualized landmarks.

## Isolation of RNA and qRT-PCR on brain tissues

Total RNA from the NAc-core, NAc-shell and HIP of JUUL Mango, JUUL Mint exposed groups, in addition to Air control group. Brain tissue was extracted with TRIzol reagent, using the manufacturer's protocol (Invitrogen, USA). The cDNA was synthesized using the iScript cDNA synthesis kit (Bio-Rad, USA). The mRNA expression level of the brain tissue was detected by qRT-PCR via iQ SYBER green I Supermix (Bio-Rad, USA) and a Bio-Rad RT-PCR instrument system. The thermocycling protocol consisted of 10 min at 95 °C, 40 cycles of 15 s at 95 °C, 30 s at 60 °C, and 20 s at 72 °C and completed with a melting curve ranging from 60°C to 95°C to facilitate distinction of specific products. A reaction with primers of *Tnfa*, *Il1b* and *Il6* was performed, the glyceraldehyde-3-phosphate dehydrogenase (GAPDH) gene was used as a housekeeping control. Data were expressed as fold change ($2^{-\Delta\Delta C_T}$) relative to the control group. The primer sequences are listed in *Table 2*.

## Brain western blot

Immunoblot assays were conducted to measure the expression of HMGB1 and RAGE proteins in the NAc core, NAc shell and HIP as described previously (*Alhaddad et al., 2014b*). Briefly, the samples were homogenized with lysis buffer containing protease and phosphatase inhibitors. The amount of protein in each tissue sample was quantified using detergent compatible protein assay (Bio-Rad, Hercules, CA, USA). Then, 10% polyacrylamide gels used, in which, an equal amount of protein from each sample was loaded. Proteins were then transferred to a PVDF membrane and blocked with 5% fat-free milk in Tris-buffered saline with Tween-20 (TBST). Membranes then incubated with appropriate primary antibodies at 4 °C (overnight): rabbit anti-HMGB1 (1:1000; Abcam), rabbit anti-RAGE (1:1000; Abcam) and mouse anti-β-tubulin (1:1000; BioLegend; used as a control loading protein). Membranes were then incubated with appropriate secondary antibody (1:5000) for 90 min at room temperature. Chemiluminescent reagents (Super Signal West Pico, Pierce Inc) were incubated with the membranes. The GeneSys imaging system was used and the digitized blot images were developed. Quantification and analysis of the expression of HMGB1, RAGE and β-tubulin were performed

using ImageJ software. Air control group data were represented as 100% to assess the change in the expression of the protein of interest as described previously (*Alhaddad et al., 2014a*).

## RNA isolation, cDNA synthesis, and qRT-PCR from cardiac tissue

RNA was isolated from samples of cells or tissue homogenized in TRIzol (Invitrogen), with subsequent extraction with chloroform, precipitation of RNA with isopropanol, and washing of RNA pellet twice with 70% ethanol. Synthesis of cDNA from isolated RNA was carried out using the High-Capacity cDNA Reverse Transcription kit with RNase inhibitor (Applied Biosystems). qRT-PCR was carried out using predesigned PrimeTime qPCR Primers (IDT) and TaqMan Universal Master Mix II with UNG (Applied Biosystems), combined with cDNA samples in a 96-well PCR plate and run on a 7,500 Fast Real–Time PCR system (Applied Biosystems). The gene expression data acquired was analyzed using the comparative $2^{-\Delta\Delta CT}$ method, with GAPDH expression levels used as the internal control.

## Cytokine/chemokine quantification from cardiac tissue

Cardiac tissue from the apices were lysed with the ProcartaPlex Cell Lysis Buffer (Invitrogen) using Miltenyi M Tubes and the Program Protein_01 of gentleMACS Dissociator (Mylteni). Protein was extracted by centrifuging at 16,000xg for 10 min at 4 °C, and supernatants were harvested. Total protein concentration was measured using the Bio-Rad DC Protein Assay Kit I, and total protein was equalized across all samples. Cytokines and chemokines were quantified using the Bio-Plex Pro Mouse Cytokine 23-plex Assay per manufacturers instructions.

## RNA isolation from and sequencing on lung tissue

Total RNA of the mice exposed toAir, Mango and Mint-flavored JUUL for 1and 3 months were extracted from homogenized whole lung tissues and preserved using Qiagen RNeasy Plus Mini Kit (Qiagen, Hilden, Germany). Four repeats were prepared for each group while samples with RNA integrity score smaller than 8.0 were excluded from the study due to poor quality. The RNA sequencing library was multiplexed and prepared using 50-base-pair single-end read with a sequencing depth of ~50 million reads per sample. Bowtie v1.3.0 was used to align the sequencing reads to the reference *Mus musculus* genome (GRCm38). Quality control of the reads and raw gene expression matrix were generated using RSEM v.1.3.0 (*Li and Dewey, 2011*). The gene expression matrix was then annotated using GENCODE (Mus_musculus.GRCm38.68.gtf). Differential gene expression analysis was conducted using the EdgeR package (*Robinson et al., 2010*; *McCarthy et al., 2012*). Genes that were sufficiently counted ( > 100 cpm) in less than two samples were excluded from the analysis to reserve the statistical power of the analysis. Differentially expressed genes were determined based on the Exact Test (*Robinson et al., 2010*; *Gibbons and Pratt, 1975*) and were further ranked based on their corresponding pi values ($\pi$). $\pi$, which considers the biological significance (log fold change) and statistical significance (p value) of a gene (*Xiao et al., 2014*), can be written as:

$$\pi = \left| LogFC \right| * \left( - \log_{10} P \right)$$

where $LogFC$ is the log fold change and $P$ is the p value of differential expression of the gene of interest. P values were then corrected using the Benjamini-Hochberg procedure to account for multiple testing. A corrected P value smaller than or equal to 0.05 was used to establish statistical significance. Biological pathway analysis and gene enrichment analysis were conducted using Ingenuity Pathway Analysis (2021; QIAGEN Redwood City). Differentially regulated canonical pathways were tabulated and ranked by their Z scores. R v4.1.0 was used for data cleaning and data visualization.

## Cytokine profiling

Cytokine and chemokine levels were assessed in the BAL with Duo-Set Enzyme-Linked Immunosorbent Assays (R&D Systems Inc, Minneapolis, MN). ELISAs were performed per manufacturer's instructions.

## Statistical analyses

Sample sizes for each group were decided at the study design phase, based on the primary outcome of neuroinflammation. Pilot data from brains of mice exposed to different e-cigarette aerosols were used to identify the n to achieve 80% power to detect a 25% change in key inflammatory proteins. Data analyses were conducted using GraphPad Prism v6.0 or v8.0. Assays with data from more than

two groups or timepoints were analyzed by two-way ANOVA with Dunnett's multiple comparisons test and are presented as means ± SEM. Groups with n < 10 were further checked for normal distribution, and if they did not pass normality testing, they were analyzed by the non-parametric Friedman test with Dunn's multiple comparisons.Quantification and analysis of Western blot protein levels of HMGB1, RAGE and β-tubulin, and histologic examination of tissue fibrosis, were performed using ImageJ software. The gene expression data acquired by qPCR was analyzed using the comparative $2^{-\Delta\Delta CT}$ method, with GAPDH (brain) and 18 S rRNA (colon) expression levels used as the internal control.

## Acknowledgements

This work was supported by grants from the National Institutes of Health (NIH), including NIH R01HL137052 (LCA) R01HL145459 (JHB), T32HL007444 (CB), American Heart Association beginning grant-in-aid 16BGIA27790079 (LCA) Postdoctoral Award 19POST34430051 (CB), UCSD grant RS169R (LCA), ATS Foundation Award for Outstanding Early Career Investigators (LCA), VA Merit Award, 1I01B × 004767 (LCA), as well as Tobacco-Related Disease Research Program grants T30IP0965 (LCA), 26IP-0040 (JHB), and 28IP-0024 (SD).

## Additional information

### Funding

| Funder | Grant reference number | Author |
|---|---|---|
| National Institutes of Health | R01HL137052 | Laura E Crotty Alexander |
| National Institutes of Health | R01HL145459 | Joan Heller Brown |
| American Heart Association | 16BGIA27790079 | Laura E Crotty Alexander |
| University of California, San Diego | RS169R | Laura E Crotty Alexander |
| American Thoracic Society | Foundation Award for Outstanding Early Career Investigators | Laura E Crotty Alexander |
| U.S. Department of Veterans Affairs | 1I01B x 004767 | Laura E Crotty Alexander |
| Tobacco-Related Disease Research Program | T30IP0965 | Laura E Crotty Alexander |
| Tobacco-Related Disease Research Program | 26IP-0040 | Joan Heller Brown |
| Tobacco-Related Disease Research Program | 28IP-0024 | Soumita Das |
| National Institutes of Health | T32HL007444 | Cameron S Brand |
| American Heart Association | Postdoctoral Award 19POST34430051 | Cameron S Brand |

The funders had no role in study design, data collection and interpretation, or the decision to submit the work for publication.

### Author contributions

Alex Moshensky, Zahoor Shah, Laura E Crotty Alexander, Conceptualization, Data curation, Formal analysis, Funding acquisition, Investigation, Methodology, Project administration, Resources, Software, Supervision, Visualization, Writing – original draft, Writing – review and editing; Cameron S Brand, Conceptualization, Data curation, Formal analysis, Funding acquisition, Project administration,

Writing – original draft, Writing – review and editing; Hasan Alhaddad, Conceptualization, Data curation, Formal analysis, Funding acquisition, Writing – original draft, Writing – review and editing; John Shin, Jorge A Masso-Silva, Ira Advani, Kenneth Park, Rita Al-Kolla, Soumita Das, Min Kwang Byun, Conceptualization, Data curation, Formal analysis, Writing – original draft, Writing – review and editing; Deepti Gunge, Data curation, Writing – original draft, Writing – review and editing; Aditi Sharma, Conceptualization, Data curation, Writing – original draft, Writing – review and editing; Sagar Mehta, Conceptualization, Data curation, Writing – review and editing; Arya Jahan, Sedtavut Nilaad, Samantha Perera, Data curation, Writing – review and editing; Jarod Olay, Conceptualization, Data curation, Formal analysis, Investigation, Writing – review and editing; Wanjun Gu, Conceptualization, Data curation, Formal analysis, Methodology, Writing – review and editing; Tatum Simonson, Data curation, Formal analysis, Methodology, Supervision, Writing – review and editing; Daniyah Almarghalani, Josephine Pham, Hoyoung Moon, Conceptualization, Data curation, Formal analysis, Writing – review and editing; Youssef Sari, Conceptualization, Data curation, Formal analysis, Methodology, Writing – original draft, Writing – review and editing; Joan Heller Brown, Conceptualization, Formal analysis, Writing – original draft, Writing – review and editing

### Author ORCIDs
Hasan Alhaddad http://orcid.org/0000-0002-3633-5705
Soumita Das http://orcid.org/0000-0003-3895-3643
Laura E Crotty Alexander http://orcid.org/0000-0002-5091-2660

### Ethics
This study was performed in strict accordance with the recommendations in the Guide for the Care and Use of Laboratory Animals of the National Institutes of Health. All of the animals were handled according to approved institutional animal care and use committee (IACUC) protocols (160204) of the University of California San Diego. All surgery was performed under ketamine and xylazine anesthesia, and every effort was made to minimize suffering.

### Decision letter and Author response
Decision letter https://doi.org/10.7554/eLife.67621.sa1
Author response https://doi.org/10.7554/eLife.67621.sa2

---

## Additional files

### Supplementary files
• Transparent reporting form

### Data availability
All data generated or analyzed during this study are included in the manuscript and supporting files.

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

## Appendix 1

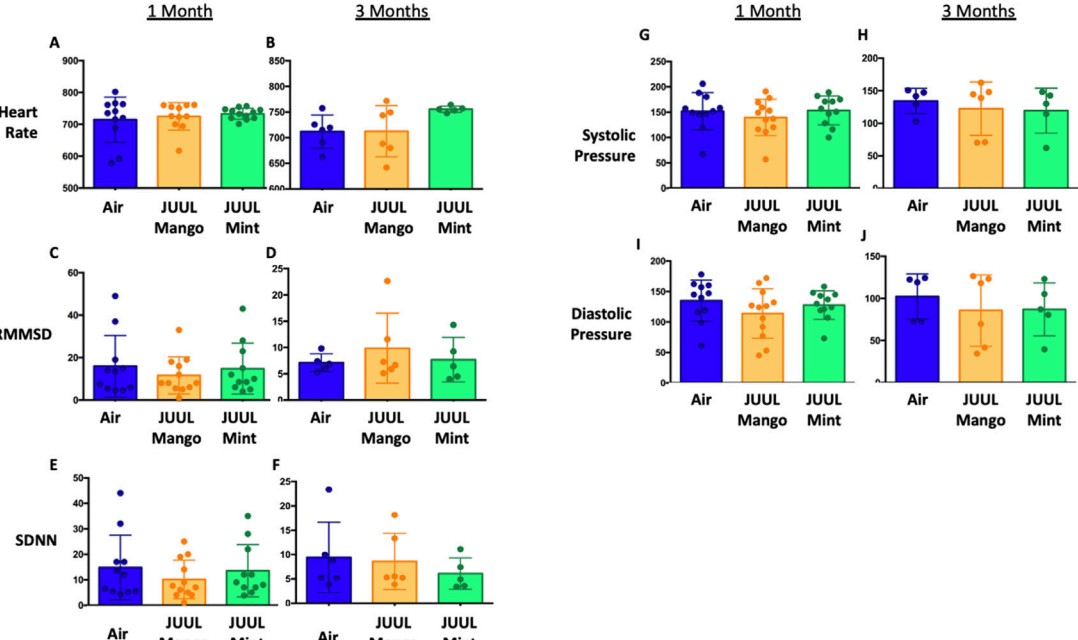

**Appendix 1—figure 1.** JUUL aerosol inhalation does not alter heart rate, heart rate variability or blood pressure. Before assessment of lung function, mice underwent heart rate and blood pressure measurements using Emka non-invasive ECG Tunnels and the CODA non-invasive blood pressure system at 1 and 3 months. Heart rate (**A–B**), heart rate variability as measured by RMMSD and SDNN (**C–F**), systolic blood pressure (**G–H**) and diastolic blood pressure (**I–J**) were unaltered by chronic e-cigarette aerosol inhalation. Data are presented as individual data points ± SEM with n = 5–11 mice per group.

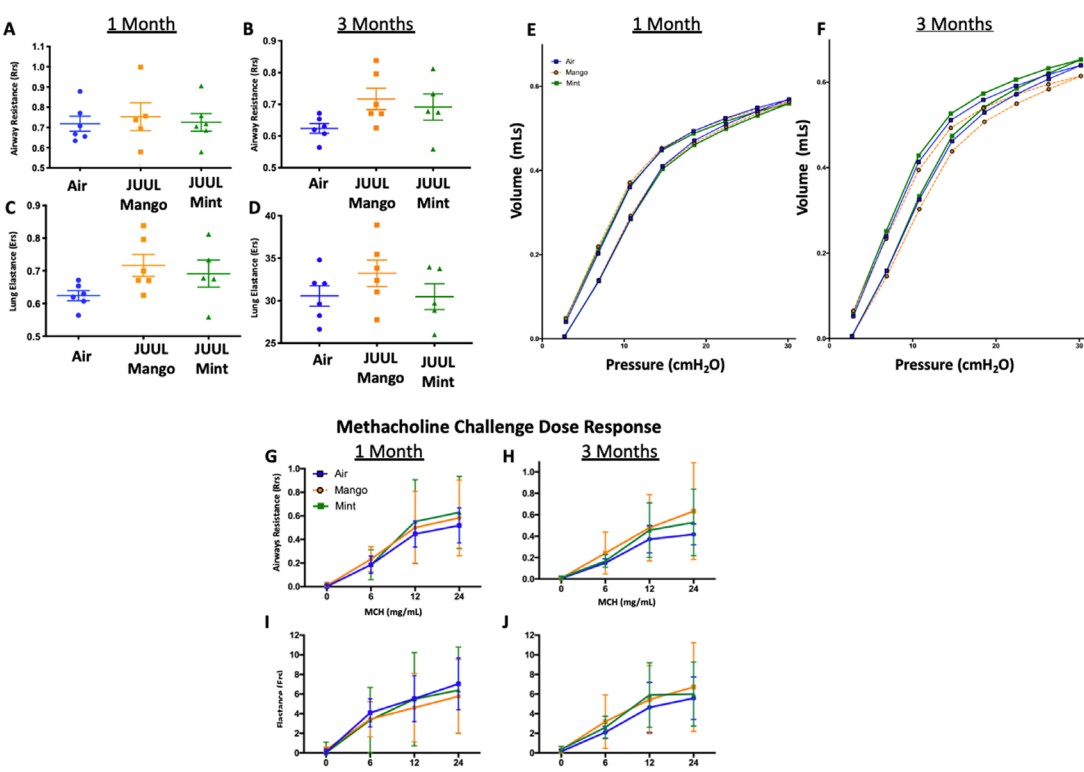

**Appendix 1—figure 2.** Chronic exposure of JUUL does not increase airways resistance or induce airways hyperreactivity. At end points prior to harvest, mice underwent tracheostomy and attached to the FlexiVent mouse ventilator (SciReq). Airways resistance, lung elastance and pressure-volume (PV) loops were measured by mechanics scans using the FlexiVent mouse ventilator (SciReq) (**A–F**), followed by the assessment of responses to methacholine (MCH) challenge at 0, 6, 12 and 24 mg/mL (**G–J**). These parameters were assessed after 1 and 3 months of exposure of JUUL. Panels show airways resistance (Rrs) at (**A**) 1 month and (**B**) 3 months, Elastance at (**C**) 1 month and (**D**) 3 months, and PV loops at (**E**) 1 month and (**F**) 3 months. There were no differences in responses to methacholine challenge by Rrs at (**G**) 1 month and (**H**) 3 months, or Ers at (**I**) 1 month and (**J**) 3 months. Data for airways resistance and lung elastance are presented as individual data points ± SEM with n = 5–11 mice per group, and PV loops as means with n = 6 mice per group.

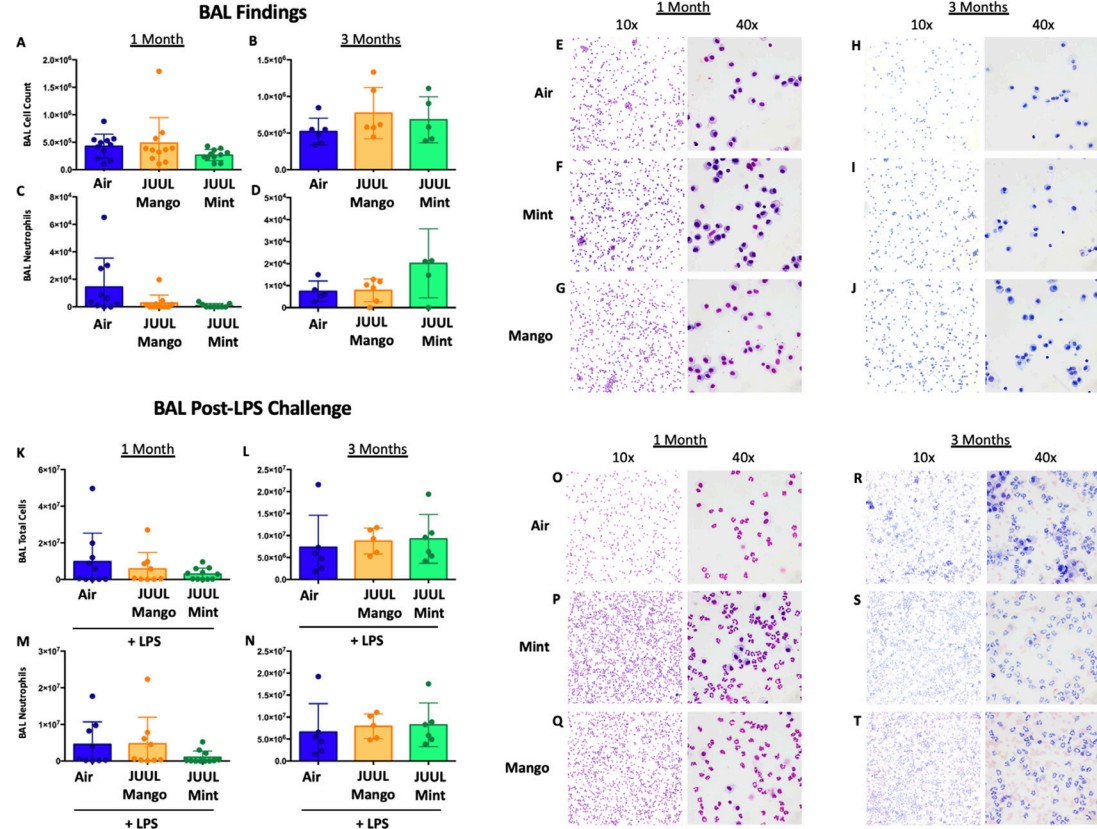

**Appendix 1—figure 3.** Chronic JUUL exposure does not affect leukocyte levels in the airways or influx into the airways and parenchyma in the setting of inhaled LPS challenge. BAL was obtained, leukocyte counts were performed. BAL total cell counts in air versus JUUL exposed mice were no different at (**A**) 1 month and (**B**) 3 months, nor were neutrophils counts at (**C**) 1 month and (**D**) 3 months. Representative pictures from Giemsa Wright stained BAL cells are shown in (**E,F,G**) for 1 month and (**H,I,J**) for 3 months. In the setting of acute lung injury induced by inhaled LPS challenge, cell counts in the BAL were no different across groups (**K–N**), with representative images from BAL cells demonstrating the same (**O–T**). Data for cell counts are presented as individual data points ± SEM with n = 5–11 mice per group.

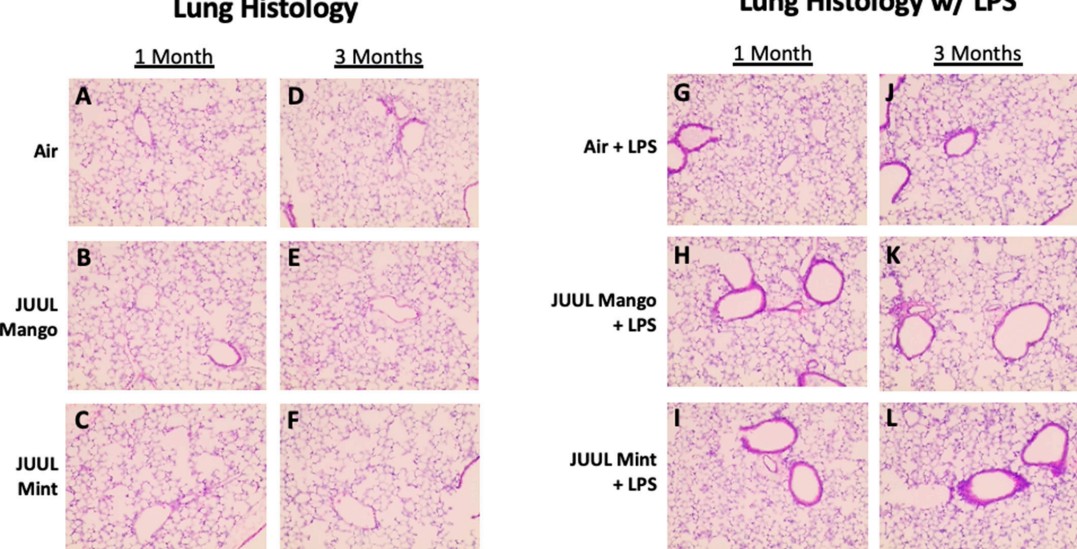

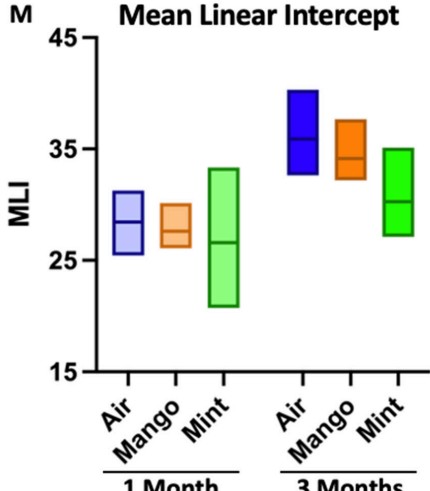

**Appendix 1—figure 4.** Chronic JUUL exposure does not affect lung parenchyma at baseline or in the setting of inhaled LPS challenge. The left lung lobe was fixed with formalin at 25 cm$^3$ water pressure and stained with H&E. Representative pictures from H&E staining of lung tissue are shown in **A,B,C** for 1 month and (**D,E,F**) for 3 months. In the setting of acute lung injury induced by inhaled LPS challenge, lung inflammation was no different by histologic evaluation (**G–L**). Mean linear intercept (MLI) was calculated for all lungs and were no different across groups at 1 and 3 months (**M**). n = 5–11 mice per group.

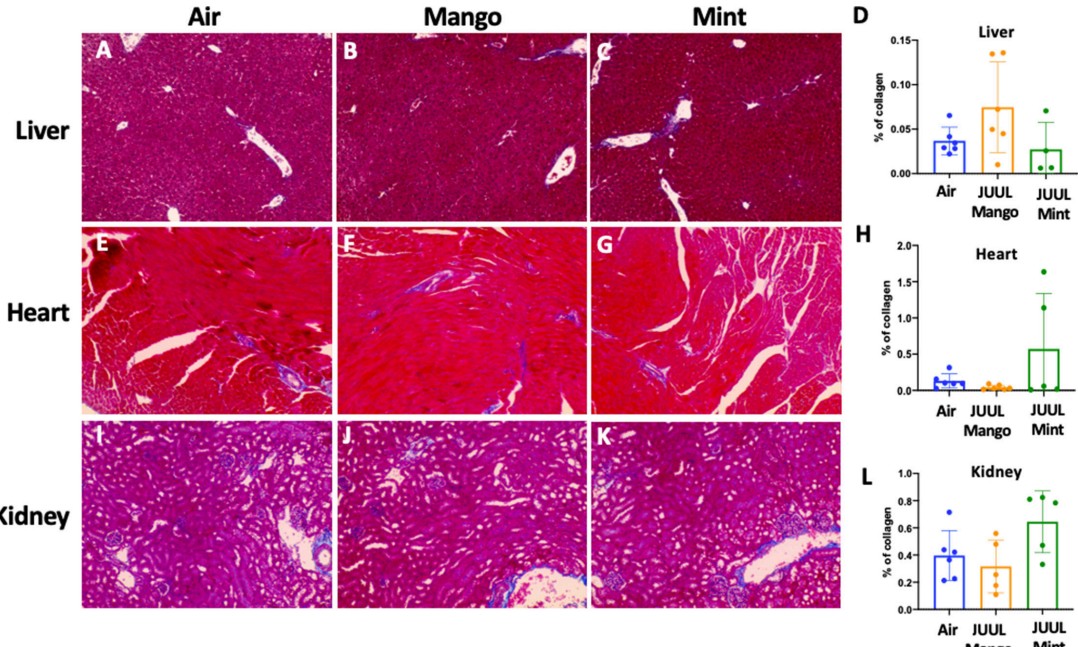

**Appendix 1—figure 5.** Chronic exposure of JUUL for 3 months does not induce fibrosis in the liver, heart, and kidney. Collagen deposition was quantified by image analysis (using ImageJ) of lung histological slides stained with Masson's trichrome. Representative pictures are shown for liver tissue in (A) Air control, (B) JUUL Mango, (C) JUUL Mint and (D) Quantification of collagen percentage in liver tissue. Representative pictures for heart tissue are shown in (E) Air control, (F) JUUL Mango, (G) JUUL Mint and (H) Quantification of collagen percentage in heart tissue. Representative pictures for heart tissue are shown in (I) Air control, (J) JUUL Mango, (K) JUUL Mint and (L) Quantification of collagen percentage in kidney tissue. Data for quantification is presented as individual data points ± SEM with n = 4–6 mice per group.

