## [Editor Report]

This study aimed to investigate the effects of vaping on inflammatory cytokine expression in multiple organs in mice. The effects of e-cigarette exposures on a different organs remain highly under investigated and, thus this study is of high interest. The manuscript has significantly improved and the authors added additional data to support their conclusion and further adapted the manuscript text and highlighted the limitations of this study.

---

## [Decision Letter]

**Decision letter after peer review:**

Thank you for submitting your article "Effect of chronic JUUL aerosol inhalation on inflammatory states of the brain, lung, heart and colon" for consideration by *eLife*. Your article has been reviewed by 3 peer reviewers, and the evaluation has been overseen by a Reviewing Editor and Paul Noble as the Senior Editor. The reviewers have opted to remain anonymous.

Essential revisions:

This is a potentially important paper that aimed to investigate the effects of vaping on multiple organs. The author show multi-organ inflammatory responses of JUUL exposure in mice. While the rationale of the current study if of high interest and timely, the manuscript in its current form remains largely descriptive and some of the conclusions are not clearly supported by the data. A major limitation is the lack of investigation of (causal) pathophysiological consequences/general organ outcomes that might be driven by the reposted inflammatory response. As such, the following revisions are required, which are further outlined below:

1. Based on the reviewer comments outlined in detail below, please revise the manuscript to avoid overstatements and include a limitations paragraph within the revised manuscript.

2. The current data are based on (mild induction of) cytokine mRNA expression. Other methods are required to confirm their expression on protein and significance.

3. Please address the potential consequences of inflammation in the distinct organs, such as the effect of neuroinflammation on animal behaviors/ psychology.

4. For the lung histology, please quantify the mean linear intercept per ATS guidelines and show representative BAL images.

5. For lung tissue, please include gene expression of cytokines.

6. If possible, please include higher n-numbers and included other tobacco or vaping device as control.

*Reviewer #1:*

This is a very interesting and potentially very important paper that shows multi-organ effects of JUUL exposure in mice. This paper is appropriate for *eLife* and should be published. I have some major comments which are listed below.

I am concerned that a lot of these studies had relatively low n numbers (n=5 in some cases) and that some of the studies may have been underpowered. Given the variability with in vivo studies, some endpoints may have been significant with more numbers. Along these lines, what is the justification for using the (parametric) ANOVA test. I'm not a statistician but I thought that the rule of thumb was that non-parametric tests should be used if n<12 since you cannot verify that the data is normally distributed. In this case, I would recommend having a statistician look at it and/or increasing some of the N's, or using the non-parametric Kruskal-Wallis test. Indeed, in some cases, the variation the variation is quite large (ie Figure 6, 7). Whilst I do not think that the low N's change the ultimate conclusions, but more rigor (ie more N's) would help solidify the paper given that it will likely be of great interest and scrutinized by the scientific community.

Figure S3. For the lung histology, please quantify the mean linear intercept per ATS guidelines and show representative BAL images.

One of the must novel conclusions from this paper is increased inflammation in the brain which the authors speculate could lead to altered moods and or change the addiction threshold. I would tend to agree with this conclusion, but could the authors perform additional mouse psychological tests to confirm this? Also, were there observable physiological responses in the vaped mice that could be reported which may correlate this conclusion, ie changes in grooming, fur ruffling or other behavioral changes?

*Reviewer #2:*

Under homeostasis conditions, the authors observed sign of inflammatory responses in the brain, the heart and the colon, while no inflammation was detected in the broncho-alveolar lavage fluid of the mice following exposures to JUUL aerosols. Also, JUUL aerosol exposures mediated airway inflammatory responses in the acute lung injury model (LPS). Further, this infection affected the inflammatory responses in the cardiac tissue. Most of the biological adverse effects induced by JUUL aerosols were flavor-specific.

Strengths include evaluating inflammation in multiple organs, as well as assessing the physiological responses in the lungs (lung function) and cardiovascular system (heart rate, blood pressure), following exposures to JUUL aerosols. Weaknesses include the fact that only female mice were used in this study. Further, the daily exposures to either air or to the JUUL aerosols lasted only 20 min per day. It is unclear how a 20-min exposure is representative of human vaping product use. Also, although daily exposures were conducted for a duration of both 1 and 3 months, time-course effects associated with JUUL aerosols are barely addressed.

Although there are a few limitations related to this study, which should be included in the manuscript, overall, the authors' claims and conclusions are based on the data that is presented through multiple figures.

– The title should include the 2 JUUL flavors that were investigated in this study.

– In the introduction and Discussion sections, the authors should consider including all in vitro and in vivo studies that have been conducted on JUUL aerosols, since there are very few of those already published articles. This would also allow to compare and contrast the results described in this manuscript.

– The authors should add a limitation paragraph at the end of the manuscript, describing the limiting factors that make the results presented here specific to this study; e.g., daily exposure duration (20 min), use of female mice only, exposures being conducted 5 days a week, etc.

– It is unclear why the authors did not evaluate the gene expression of various cytokines in the lung tissue.

– The authors should state the rationale for using only female mice in this study.

– It is unclear why the authors do not present their gene expression data in the form of a heatmap instead of 24 small individual graphs (one graph per gene) (e.g., Figures 3 and 6).

*Reviewer #3:*

In this study, Alex Moshensky et al., investigated effects of chronic aerosol inhalation of flavored JUUL on inflammatory markers in several organs, including brain, lung, heart, and colon in a mouse model. They found that JUUL inhalation upregulated a number of cytokine and chemokine gene expression and increased HMGB1 and RAGE in the nucleus accumbens. Inflammatory gene expression increased in colon, and cardiopulmonary inflammatory responses to acute lung injury with LPS were exacerbated in the heart. They also found flavor-dependent changes in several responses.

Overall, it is a descriptive study and the conclusions was not clearly supported by the data.

Strengths

Due to the rapid evolution of vaping devices, the data on health effects of Pod devices are scarce. This study provides useful information on the inflammatory change caused by chronic JUUL aerosol inhalation.

Weaknesses

1. The authors observed neuroinflammation in brain regions responsible for behavior modification, drug reward and formation of anxious or depressive behaviors after exposure to JUUL. The importance of the neuroinflammation is still unclear. It would help demonstrate the pathogenic role of the neuroinflammation by testing animal behaviors. Similar issue for other organ inflammation.

2. Majority of the data are inflammatory cytokine mRNA expression. Other methods would be needed to confirm their expression.

3. The author seemed to assume the difference between JUUL Mango and JUUL Mint is flavor and then came up with the conclusion regarding flavor-dependent changes in several inflammatory responses. Evidence is needed to approve the assumption.

4. In most cases, the change of inflammatory cytokines is mild ~2 fold. The author should demonstrate how these marginal change could affect pathophysiology.

5. To fully evaluate the health impact of evolving cigarette, it would be informative to included other tobacco or vaping device as control.

6. The longest exposure in the study is 3 months. It is not convicting to come up with conclusions regarding chronic exposure. Some organ showing no difference may be due to the timing.

7. It would help reader to see the dynamic change of inflammation by combining 1 month with 3 month data into one graph for individual cytokines.

8. ELISA or western blot should be included to measure cytokine for two reasons – the fold-change is mild for most cytokines and difference in mRNA level does not necessarily mean difference in protein.

---

## [Author Response]

Essential revisions:This is a potentially important paper that aimed to investigate the effects of vaping on multiple organs. The author show multi-organ inflammatory responses of JUUL exposure in mice. While the rationale of the current study if of high interest and timely, the manuscript in its current form remains largely descriptive and some of the conclusions are not clearly supported by the data. A major limitation is the lack of investigation of (causal) pathophysiological consequences/general organ outcomes that might be driven by the reposted inflammatory response. As such, the following revisions are required, which are further outlined below:

Thank you for the detailed review of this original research manuscript. We appreciate the reviewer identifying it as “of high interest and timely.” Of note, part of working in a new field such as e-cigarettes involves assessing for and detecting signals of potential clinical relevance. We do not mean to overstate our findings and have revised the manuscript to avoid that.

1. Based on the reviewer comments outlined in detail below, please revise the manuscript to avoid overstatements and include a limitations paragraph within the revised manuscript.

We have gone through the manuscript in detail and have altered the verbiage in multiple places to try to ensure that our findings are not overstated. Also, we have separated out our existing limitations within the discussion as a distinct section to identify the limitations of our study more readily, and have included additional points to more comprehensively discuss the limitations of our study, leading to this section being 3 full paragraphs (see pages 18 to 19).

2. The current data are based on (mild induction of) cytokine mRNA expression. Other methods are required to confirm their expression on protein and significance.

Of note, in the original submission, we included protein quantification data for both the brain (Figure 2) and the lung (Figure 7). We have taken the reviewers comments to heart and have conducted protein-level assays on the cardiac tissues as well, yielding additional data (*new Figure 4*) that has been added to the methods, results, figures and discussion. Unfortunately, we do not have any additional colonic tissue for protein-level assessments, as all of the tissue was used for the gene transcription (Figure 5) and histologic studies. But to take a step back, these studies were originally intended to examine the broad reaching impact of e-cigarette aerosols across the body. This work, and thus this manuscript, was designed to highlight changes at the gene expression level, to demonstrate that e-cigarette use is not benign and does have broad-reaching effects on gene expression. We agree that more work is needed to fully define the impact of e-cigarette use at the protein, cellular, and organ level, but the majority of that work is beyond the scope of this manuscript. To bring the focus back to gene expression, we have conducted RNAseq on the lungs of JUUL exposed mice, and have included those data herein (*new Figure 6*) to highlight the effects of e-cigarette aerosols on gene expression in the lung, with a particular focus on differences between Mint and Mango flavors (the most popular JUUL flavors at the time of this study). Methods, results, and discussion related to the RNAseq results have been added to the manuscript. These data support the hypothesis that e-cigarette aerosol inhalation fundamentally alters the lung, which raises the specter of downstream health effects.

3. Please address the potential consequences of inflammation in the distinct organs, such as the effect of neuroinflammation on animal behaviors/ psychology.

We are thrilled that the Reviewer is as interested in these implications as we are, because we believe the neuroinflammation detected is quite frightening, particularly because it is likely to impact both behavior and mood. We have added further discussion regarding the potential consequences of inflammation in each of the organs (pages 13-18), with an emphasis on the effects of neuroinflammation on behavior and psychology. We have subdivided the Discussion section to highlight potential effects on each distinct organ. It is clear that further studies are needed to best assess potential psychological and behavioral changes induced by e-cigarette aerosol inhalation. Of note, we did not observe any overt behavioral changes – we closely observe the mice both during and after exposures and make notes regarding grouping, fur, and activity level – none of which were changed by the different vaping exposures. We have added the lack of dedicated behavioral and psychological evaluations as a limitation of this work and as an opportunity for discovery in future studies (page 19).

4. For the lung histology, please quantify the mean linear intercept per ATS guidelines and show representative BAL images.

We have conducted the mean linear intercept (MLI) measurements on e-cigarette aerosol exposed lungs and controls per ATS guidelines and have added these data to the manuscript (*new Appendix 1- Figure 4M*). We paired these data with the original histology images (Appendix 1- Figure 4A-4L). We have added appropriate methods (pages 22-23) and results (page 9) as well. Of note, the MLI data matches our original physiologic assessments of lung function (Appendix 1 – Figure 2A-2J), including elastance and compliance, which are known to change in the setting of emphysema. MLI, lung elastance and compliance were no different across inhalant groups and controls. Further, we have taken representative images of Giemsa Wright stained BAL samples, and have added these to the manuscript (*new Appendix 1 Figure 3E-3J and 3O-3T*, paired with BAL cell count data).

5. For lung tissue, please include gene expression of cytokines.

Because the lung is the first in line to interact and respond to e-cigarette aerosols, and because it is designed to protect itself and the rest of the body against toxic inhalants, we hypothesized that gene expression in this tissue would be altered in complex patterns – with evidence of both increased expression of some inflammatory genes while others are downregulated. To obtain the most broad and detailed data possible, we assessed gene transcription across the lung via RNAseq (*new Figure 6*). We have added these RNAseq data, with relevant additions to the Abstract, Methods, Results and Discussion sections.

6. If possible, please include higher n-numbers and included other tobacco or vaping device as control.

We conducted power analyses prior to the start of the studies to identify the number of animals per group to use, based on our past studies of inflammatory changes induced by inhalants, infections and asthma. We set the target number of mice (n) at that time, such that these studies would be powered to detect a 25% change in cytokine expression. Based on discussions with our biostatisticians, we came to the conclusion that it would not be statistically appropriate to run more mice to increase the n when our primary outcome remains the same. In addition, and most relevant to this specific recommendation, JUUL Mint and JUUL Mango flavors are no longer on the market, such that extensive further studies are not feasible. While these two flavors are not available anymore, they were composed of an array of chemicals commonly found in other flavors (but in different combinations), such that we believe that these data are most likely relevant to other vapes. In particular, JUUL Mint shares chemical features with JUUL Menthol, which took its place as one of the most popular JUUL flavors. The discontinuation of these flavors has been added as a limitation within the Discussion (page 19).

Reviewer #1:This is a very interesting and potentially very important paper that shows multi-organ effects of JUUL exposure in mice. This paper is appropriate for eLife and should be published. I have some major comments which are listed below.

Thank you so much for your comments, we worked hard on these studies and are excited to share the novel data generated.

I am concerned that a lot of these studies had relatively low n numbers (n=5 in some cases) and that some of the studies may have been underpowered. Given the variability with in vivo studies, some endpoints may have been significant with more numbers. Along these lines, what is the justification for using the (parametric) ANOVA test. I'm not a statistician but I thought that the rule of thumb was that non-parametric tests should be used if n<12 since you cannot verify that the data is normally distributed. In this case, I would recommend having a statistician look at it and/or increasing some of the N's, or using the non-parametric Kruskal-Wallis test. Indeed, in some cases, the variation the variation is quite large (ie Figure 6, 7). Whilst I do not think that the low N's change the ultimate conclusions, but more rigor (ie more N's) would help solidify the paper given that it will likely be of great interest and scrutinized by the scientific community.

We conducted power analyses prior to the start of the studies to identify the number of animals per group to use, based on our past studies of inflammatory changes induced by inhalants, infections and asthma. We set the target number of mice (n) at that time, such that these studies would be powered to detect a 25% change in cytokine expression. We did go through and reviewed all of the data with our biostatisticians, we came to the conclusion that it would not be statistically appropriate to run more mice to increase the n when our primary outcome remains the same. We double-checked that the ANOVAs with corrections for multiple comparisons were correct for each particular experiment. Discussion with our statistician confirmed that ANOVA is correct as long as the data passed normality testing, which was done. An additional point, and most relevant to this specific recommendation, JUUL Mint and JUUL Mango flavors are no longer on the market, such that extensive further studies are not feasible. While these two flavors are not available anymore, they were composed of an array of chemicals commonly found in other flavors (but in different combinations), such that we believe that these data are most likely relevant to other vapes. In particular, JUUL Mint shares chemical features with JUUL Menthol, which took its place as one of the most popular JUUL flavors. The discontinuation of these flavors has been added as a limitation within the Discussion (page 19).

Figure S3. For the lung histology, please quantify the mean linear intercept per ATS guidelines and show representative BAL images.

We have conducted the mean linear intercept (MLI) measurements on e-cigarette aerosol exposed lungs and controls per ATS guidelines and have added these data to the manuscript (*new Appendix 1- Figure 4M*). We paired these data with the original histology images (Appendix 1 – Figure 4A-4L). We have added appropriate methods (pages 21-22) and results (page 9) as well. Of note, the MLI data matches our original physiologic assessments of lung function (Appendix 1 – Figure 2A-2J), including elastance and compliance, which are known to change in the setting of emphysema. MLI, lung elastance and compliance were no different across inhalant groups and controls. Further, we have taken representative images of Giemsa Wright stained BAL samples, and have added these to the manuscript (*new* Appendix 1 Figure – 3E-3J and 3O-3T) paired with BAL cell count data.

One of the must novel conclusions from this paper is increased inflammation in the brain which the authors speculate could lead to altered moods and or change the addiction threshold. I would tend to agree with this conclusion, but could the authors perform additional mouse psychological tests to confirm this? Also, were there observable physiological responses in the vaped mice that could be reported which may correlate this conclusion, ie changes in grooming, fur ruffling or other behavioral changes?

We are thrilled that the Reviewer is as interested in these implications as we are, because we believe the neuroinflammation detected is quite frightening, particularly because it is likely to impact both behavior and mood. We have added further discussion regarding the potential consequences of inflammation in each of the organs (pages 13-19), with an emphasis on the effects of neuroinflammation on behavior and psychology. We have subdivided the Discussion section to highlight potential effects on each distinct organ.

While we are not a behavioral lab, and thus running behavioral studies in mice is beyond the scope of both our lab and this manuscript, we agree that the neuroinflammation is of great interest and further studies are needed to best assess potential psychological and behavioral changes. Of note, we did not observe any overt behavioral changes – we closely observe the mice both during and after exposures and make notes regarding grouping, fur, and activity level – none of which were changed by the different vaping exposures. We have added the lack of dedicated behavioral and psychological evaluations as a limitation of this work and as an opportunity for discovery in future studies (page 19-20).

Reviewer #2:Under homeostasis conditions, the authors observed sign of inflammatory responses in the brain, the heart and the colon, while no inflammation was detected in the broncho-alveolar lavage fluid of the mice following exposures to JUUL aerosols. Also, JUUL aerosol exposures mediated airway inflammatory responses in the acute lung injury model (LPS). Further, this infection affected the inflammatory responses in the cardiac tissue. Most of the biological adverse effects induced by JUUL aerosols were flavor-specific.Strengths include evaluating inflammation in multiple organs, as well as assessing the physiological responses in the lungs (lung function) and cardiovascular system (heart rate, blood pressure), following exposures to JUUL aerosols. Weaknesses include the fact that only female mice were used in this study. Further, the daily exposures to either air or to the JUUL aerosols lasted only 20 min per day. It is unclear how a 20-min exposure is representative of human vaping product use. Also, although daily exposures were conducted for a duration of both 1 and 3 months, time-course effects associated with JUUL aerosols are barely addressed.

We would like to thank the reviewer for their positive comments on our manuscript. We apologize for our error; in reality we exposed mice for 20 minutes three times daily, so one hour in total per day. We have corrected this error within our Methods. We designed the exposures this way to better mimic human e-cigarette use throughout the day (instead in one intense vaping session per day, which is not the norm). We agree that there is a limitation in using only female mice in the study in case that there are sex-dependent effects, which is definitely an interesting question. We typically start with one sex of mice and then run repeat experiments with the other sex. Unfortunately, this study faced problems beyond our control that prevented us from performing further experiments. In late 2019 the FDA was moving to ban specific flavors for pod devices, which include those for Mint and Mango. In anticipation of the new regulations, JUUL ultimately decided to discontinue JUUL Mint and Mango, and soon they were out of the market. The same process occurred with the other popular JUUL flavors such as Crème Brûlée and Cucumber. We have expanded the limitation section within the Discussion, and have pointed out that because these studies were conducted in female mice alone, the results may not represent effects in males.

Although there are a few limitations related to this study, which should be included in the manuscript, overall, the authors' claims and conclusions are based on the data that is presented through multiple figures.

We appreciate the Reviewers comments and have added limitations about the study size, power, lack of male subjects, etc. to the Discussion section.

– The title should include the 2 JUUL flavors that were investigated in this study.

We have added the 2 flavors to the title of the study. Also, we swapped out the brand name JUUL for the type of e-cigarette / e-device used and have removed the term chronic. The new title is now: “Effects of Mango and Mint pod-based e-cigarette aerosol inhalation on inflammatory states of the brain, lung, heart and colon in mice”

– In the introduction and Discussion sections, the authors should consider including all in vitro and in vivo studies that have been conducted on JUUL aerosols, since there are very few of those already published articles. This would also allow to compare and contrast the results described in this manuscript.

We conducted a review of the literature to more comprehensively include other JUUL studies done to date. The following key words were inserted into PubMed: “in vitro and JUUL aerosol”, “in vivo and JUUL aerosol”, and “JUUL aerosol”. The first search, “in vitro and JUUL aerosol”, had 7 results of which 4 were relevant. The second search “in vivo and JUUL aerosol” had in 1 result; however, the same study was already included within the former search. The final broad search “JUUL aerosol” had 45 results, of which 9 (not including duplicates from the previous searches) were relevant. 1 article had already been cited. The literature review found a total of 13 articles all of which have been cited in the revised manuscript, primarily via the following paragraph added to the Introduction:

Introduction: “Studies of JUUL to date have been predominantly subacute and acute exposures with a focus on in vitro and in vivo experiments. A majority of the current literature examined the cytotoxic effects of varying JUUL flavors (Crème Brulee, Cool Mint, Fruit Medley, Tobacco, Menthol, etc.) on cells of the respiratory system (10-17). Most in vitro studies concluded that JUUL aerosols are cytotoxic and impair cell function (10, 12-16). Additional studies defined the chemical profiles of JUUL flavored cartridges (11) or examined aerosol emission and oxidant yields from flavored JUUL pods (18, 19). These studies concluded that while JUUL pods had significantly lower oxidant yields in comparison to combustible cigarettes, the nicotine concentrations were substantially greater (18-20). Another study assessed whether exposure to JUUL aerosol (in comparison to aerosol containing Vitamin E Acetate) played a causal role in EVALI patients (21). While the study revealed that 15 days of exposure to JUUL aerosol did not cause direct lung injury, the authors cautioned that chronic exposure to nicotine may still have a disruptive effect on lung physiology (21). While these studies have laid the groundwork for assessing the acute impact of JUUL use on the respiratory system, much work remains to be done on the effects across the body.”

– The authors should add a limitation paragraph at the end of the manuscript, describing the limiting factors that make the results presented here specific to this study; e.g., daily exposure duration (20 min), use of female mice only, exposures being conducted 5 days a week, etc.

We apologize for our error; in reality we exposed mice for 20 minutes three times daily, so one hour in total per day. We have corrected this error within our Methods. We designed the exposures this way to better mimic human e-cigarette use throughout the day (instead in one intense vaping session per day, which is not the norm). We have expanded the limitation section within the Discussion to include sex, duration of exposure (sub-acute, chronic, and long-term), and weekday only exposures. This limitations section is specifically labeled to allow easy identification (pages 18-19).

– It is unclear why the authors did not evaluate the gene expression of various cytokines in the lung tissue.

We agree that these data are highly relevant and could strengthen our conclusion. Originally, we did not expect to find major differences in inflammation at the time points that we assessed since there were no differences in lung physiology and histopathology, or in cellular recruitment to the airways. But because many changes could be occurring at the gene expression level, that could set the stage for pathologic responses to common clinical challenges, we conducted new RNAseq studies on lung tissues and have included these data as *new Figure 6*. We have added relevant information to the abstract, methods, results and Discussion sections as well.

– The authors should state the rationale for using only female mice in this study.

In late 2019, in anticipation of the new regulations, JUUL ultimately decided to discontinue popular flavorants such as Mint and Mango. As a result, we were unable to replicate the study in male mice since the flavorants are no longer on the market. We have previously found different results when using male versus female mice in our inhalant exposure experiments, and typically run one sex followed by the other (to prevent inter-mixing during the 3-times daily exposures). We have added further discussion regarding this limitation into the discussion. We do plan on running JUUL exposures in male mice, but these will be limited to Tobacco and Menthol flavors since these are currently the only ones available on the market.

– It is unclear why the authors do not present their gene expression data in the form of a heatmap instead of 24 small individual graphs (one graph per gene) (e.g., Figures 3 and 6).

We understand that figure does look a bit busy, however we believe that it is important to show the variability within each experimental group (each mouse is represented as an individual dot) instead of only showing the mean that a heat map would show. This style of data presentation also allows error bars to be included.

Reviewer #3:In this study, Alex Moshensky et al., investigated effects of chronic aerosol inhalation of flavored JUUL on inflammatory markers in several organs, including brain, lung, heart, and colon in a mouse model. They found that JUUL inhalation upregulated a number of cytokine and chemokine gene expression and increased HMGB1 and RAGE in the nucleus accumbens. Inflammatory gene expression increased in colon, and cardiopulmonary inflammatory responses to acute lung injury with LPS were exacerbated in the heart. They also found flavor-dependent changes in several responses.Overall, it is a descriptive study and the conclusions was not clearly supported by the data.StrengthsDue to the rapid evolution of vaping devices, the data on health effects of Pod devices are scarce. This study provides useful information on the inflammatory change caused by chronic JUUL aerosol inhalation.

We would like to thank the reviewer for identifying the novelty of our data within the e-cigarette field and beyond, as almost nothing is known about pod based devices.

Weaknesses1. The authors observed neuroinflammation in brain regions responsible for behavior modification, drug reward and formation of anxious or depressive behaviors after exposure to JUUL. The importance of the neuroinflammation is still unclear. It would help demonstrate the pathogenic role of the neuroinflammation by testing animal behaviors. Similar issue for other organ inflammation.

We are an immunology, inflammation, and lung physiology lab, thus, behavioral studies are beyond the scope of both our lab and this manuscript. However, we agree that the neuroinflammation is of great interest and is highly likely to impact behavior and mood. Further studies are needed to best assess potential psychological and behavioral changes. We believe this work is important to share such that dedicated behavioral science labs can undertake these important studies. We have added these important limitations to the discussion.

2. Majority of the data are inflammatory cytokine mRNA expression. Other methods would be needed to confirm their expression.

Of note, in the original submission, we included protein quantification data for both the brain and the lung. We have taken the reviewers comments to heart and have conducted protein-level assays on the cardiac tissues as well, yielding additional data (new Figure 4) that has been added to the methods, results, figures and discussion. Unfortunately, we do not have any additional colonic tissue for protein-level assessments, as all of the tissue was used for the gene transcription and histologic studies. But to take a step back, these studies were originally intended to examine the broad reaching impact of e-cigarette aerosols across the body. This work, and thus this manuscript, was designed to highlight changes at the gene expression level, to demonstrate that e-cigarette use is not benign and does have broad-reaching effects on gene expression. We agree that more work is needed to fully define the impact of e-cigarette use at the protein, cellular, and organ level, but the majority of that work is beyond the scope of this manuscript. To bring the focus back to gene expression, we have conducted RNAseq on the lungs of JUUL exposed mice, and have included those data herein to highlight the effects of e-cigarette aerosols on gene expression in the lung, with a particular focus on differences between Mint and Mango flavors (the most popular JUUL flavors at the time of this study). These new data (*new Figure 6*) support the hypothesis that e-cigarette aerosol inhalation fundamentally alters the lung, which raises the specter of downstream health effects.

3. The author seemed to assume the difference between JUUL Mango and JUUL Mint is flavor and then came up with the conclusion regarding flavor-dependent changes in several inflammatory responses. Evidence is needed to approve the assumption.

Although the formulation of JUUL e-liquids is proprietary, their website claims simplicity (https://www.juul.com/learn/pods) in that they use pharmaceutical grade propylene glycol and glycerol (which makes up the majority of their e-liquids), in order to form an aerosol which carries pharmaceutical grade nicotine and benzoic acid (when combined, create a nicotine salt), and flavors (which can be can be a mixture of natural and artificial ingredients). Thus, according to their website the only difference among the different JUUL pods would be the flavoring components. Hence, we concluded that differences observed in our study between Mint vs Mango should be most likely due to flavor-dependent effects, since base components should be the same. To support this flavor-dependent effect, a study from Omaiye et al., in 2019 (PMID: 30896936) showed the variety of different flavoring chemical in all JUUL flavors and how the different JUUL vapors induce different level of cytotoxicity in BEAS-2B cells in vitro based their flavor. We have added relevant discussion to the manuscript.

4. In most cases, the change of inflammatory cytokines is mild ~2 fold. The author should demonstrate how these marginal change could affect pathophysiology.

We agree with the reviewer that the majority of changes in cytokines were relatively small. However, the fact that multiple cytokines are changing in concert indicates a significant shift in immunophenotyping across organs. We are most concerned about how these shifts in the inflammatory state will alter an e-cigarette vapers response to common clinical challenges. In Dr. Kheradmand’s recent work, mice exposed to e-cigarette aerosols with and without nicotine were much more susceptible to acute lung injury in the setting of viral pneumonia. In our work, we utilized the LPS model of acute lung injury to take a first look at the potential impact of JUUL inhalation in particular on susceptibility to lung inflammation. Further work is needed to truly define how the subtle, broad shifts in the cytokine milieu across organs will impact the health of e-cigarette vapers. We have added relevant discussion to the manuscript (page 20):

“While the majority of changes in cytokine levels induced by e-cigarette exposure across organ systems were relatively small, the fact that multiple cytokines changed in concert indicates a significant shift in immunophenotyping across organs. We are most concerned about how these shifts in the inflammatory state will alter an e-cigarette users response to common clinical challenges. For example, Madison *et al.,* exposed mice to e-cigarette aerosols with and without nicotine and found that both exposures increased susceptibility to acute lung injury in the setting of viral pneumonia. In our work, we utilized the LPS model of acute lung injury to take a first look at the potential impact of JUUL inhalation in particular on susceptibility to lung inflammation. Further work is needed to truly define how the subtle, broad shifts in the cytokine milieu across organs will impact the health of e-cigarette vapers.”

5. To fully evaluate the health impact of evolving cigarette, it would be informative to included other tobacco or vaping device as control.

We agree that such comparisons are likely to provide insight into the differences between devices and formulations and versus cigarette smoke, and thus will be incredibly important for the field. However, these comparisons were beyond the scope of this study, whose main goal was to assess the inflammatory and physiological aspects of JUUL in particular. We believe this to be important because JUUL e-cigarettes are the most popular of all e-cigarette devices, and many young users do not use other e-devices or conventional tobacco. Thus, our primary objective of this work was to specifically assess the safety or risk of this device in particular (versus not using any inhalant at all). However, because we have run parallel studies in the past with vape pens, box mods, and conventional tobacco, we are hopeful to start combining data to look for trends and differences across inhalant exposures. For example, we recently published our work on differences in metabolites in the circulation of mice exposed to a wide variety of e-cigarette based inhalants (Moshensky et al., Vaping induced metabolomic signatures in the circulation of mice are driven by device type, eliquid, exposure duration and sex).

ERJ Open. July 2021 PMID: 34262972. This study is one of the few studies that have employed animal models to test JUUL devices and the only one assessing their effects in different organs, and although we agree that comparisons with other devices is important, it was not the goal of this study.

6. The longest exposure in the study is 3 months. It is not convicting to come up with conclusions regarding chronic exposure. Some organ showing no difference may be due to the timing.

We have altered the wording throughout the manuscript to clarify that the 3-month duration is equivalent to 10 to 20 years of inhalant use versus 40 to 50 years for a 6 to 12 month model. We have also removed many instances of the descriptive terms acute, sub-acute and chronic across the manuscript, as focused on using the absolute duration of exposure instead, to avoid accidental extrapolation to longer exposures. Because we utilized cellular and molecular based assays, we were not relying on identifying organ level pathology such as fibrosis, emphysema, and organ dysfunction, all of which would require longer exposures.

7. It would help reader to see the dynamic change of inflammation by combining 1 month with 3 month data into one graph for individual cytokines.

We have generated these new graphs and have included them as Figure supplements: Figure 1 —figure supplement 1, Figure 2 —figure supplement 1, Figure 3 —figure supplement 1, Figure 5 —figure supplement 1, Figure 7 —figure supplement 1, Figure 8 —figure supplement 1, Figure 9 —figure supplement 1.

8. ELISA or western blot should be included to measure cytokine for two reasons – the fold-change is mild for most cytokines and difference in mRNA level does not necessarily mean difference in protein.

We agree with the reviewer, that confirmation with protein levels would yield the strongest data possible. In the original manuscript we did include protein-based quantification data for the brain and lung. We have now run protein-based assays for the cardiac tissue (*new Figure 4*). However, we did not have enough tissue to conduct protein-based studies on colon tissue. Although we were unable to include protein based quantification for all organs, we do feel that our original intention of demonstrating broad reaching effects of e-cigarette aerosols across the body was still achieved. But we absolutely agree that more work is needed to fully define the impact of e-cigarette use at the protein, cellular, and organ level.